# A single microRNA miR-195 rescues the arrested B cell development induced by EBF1 deficiency

Yuji Miyatake[1†], Takeshi Kamakura[2†], Tomokatsu Ikawa[3], Ryo Yanagiya[1], Ryutaro Kotaki[1], Kazuaki Kameda[1], Ryo Koyama Nasu[4,5], Kazuki Okuyama[6], Ken-ichi Hirano[7], Hiroyuki Hosokawa[7], Katsuto Hozumi[7], Masato Ohtsuka[8], Takahiro Kisikawa[9], Chikako Shibata[9], Motoyuki Otsuka[10], Reo Maruyama[11], Kiyoshi Ando[12], Tomohiro Kurosaki[13,14,15], Hiroshi Kawamoto[16], Ai Kotani[1,2]*

[1]Department of Innovative Medical Science, Tokai University School of Medicine, Isehara, Japan; [2]Department of Regulation of Infectious Cancer, Research Institute for Microbial Diseases, The University of Osaka, Osaka, Japan; [3]Division of Immunology and Allergy, Research Institute for Biomedical Sciences, Tokyo University of Science, Noda, Japan; [4]Department of Immunology, Graduate School of Medicine, Chiba University, Chiba, Japan; [5]Department of Experimental Immunology, Graduate School of Medicine, Chiba University, Chiba, Japan; [6]Laboratory for Transcriptional Regulation, RIKEN Center for Integrative Medical Sciences, Yokohama, Japan; [7]Department of Immunology, Tokai University School of Medicine, Isehara, Japan; [8]Department of Molecular Life Science, Tokai University School of Medicine, Isehara, Japan; [9]Department of Gastroenterology, Graduate School of Medicine, The University of Tokyo, Tokyo, Japan; [10]Department of Gastroenterology and Hepatology, Okayama University Hospital, Okayama, Japan; [11]Project for Cancer Epigenomics, Cancer Institute, Japanese Foundation for Cancer Research, Tokyo, Japan; [12]Hematology and Oncology, Research Institute for Radiation Biology and Medicine, Hiroshima University, Hiroshima, Japan; [13]Laboratory of Lymphocyte Differentiation, WPI Immunology Frontier Research Center, Osaka University, Osaka, Japan; [14]Laboratory for Infectious Disease Education and Research, Osaka University, Osaka, Japan; [15]Laboratory for Lymphocyte Differentiation, RIKEN Center for Integrative Medical Sciences, Yokohama, Japan; [16]Laboratory of Immunology, Institute for Life and Medical Sciences, Kyoto University, Kyoto, Japan

*For correspondence:
aikotani@k-lab.jp

†These authors contributed equally to this work

## eLife Assessment

This **useful** study reports that the exogenous expression of the microRNA miR-195 can partially compensate in early B cell development for the loss of EBF1, one of the key transcription factors in B cells. While this finding will be of interest to those studying lymphocyte development, the evidence, particularly with regard to the molecular mechanisms that underpin the effect of miR-195, is currently **incomplete**.

**Abstract** Accumulated studies have reported that hematopoietic differentiation was primarily regulated by transcription factors. Early B cell factor 1 (EBF1) is an essential transcription factor for B lymphopoiesis. Contrary to the canonical notion, we found that a single miRNA, miRNA-195 (*Mir195*) transduction let *Ebf1*-deficient hematopoietic progenitor cells (HPCs) express CD19, carry

out V(D)J recombination and class switch recombination, which implied that B cell matured without EBF1. A part of the mechanism was caused by FOXO1 accumulation via inhibition of FOXO1 phosphorylation pathways in which targets of *Mir195* are enriched. These results suggested that some miRNA transductions could function as alternatives to transcription factors.

## Introduction

Developmental hierarchy in hematopoiesis has been widely researched, and it is well known that proper stimulation leads hematopoietic stem cells (HPCs) into B cell lineage. Lineage specification is primarily regulated at the transcriptional level, thus lineage-specific transcription factors are considered to be indispensable for differentiation (*Zhu and Emerson, 2002*; *Ikawa et al., 2004*). B cell development requires multiple transcription factors, especially Early B cell Factor 1 (EBF1), Paired Box 5 (Pax5), and E2A. Pax5 and E2A are critical transcription factors for early B cell development, but they cannot rescue *Ebf1*-deficient HPCs from failure of B cell lineage commitment (*Lin et al., 2010*). Conversely, ectopic expression of EBF1 is able to rescue Pax5, E2A, and PU.1 deleted progenitor cells from B lymphopoiesis arrest, and thus EBF1 is considered more potent than the other transcription factors (*Ikawa et al., 2004*; *Pongubala et al., 2008*; *Györy et al., 2012*). As the most potent transcription factor, EBF1 is essential for pre-pro-B cell to become pro-B cell; namely, *Ebf1*$^{-/-}$ cell expresses B220 but is disabled to express CD19 (*Medina et al., 2004*).

MicroRNAs (miRNAs) are small noncoding RNA containing approximately 22 nucleotides that regulate several target protein expressions mediating deadenylation and translation by posttranscriptionally repressing or decaying target messenger RNAs (mRNAs) (*Bartel, 2009*; *Carthew and Sontheimer, 2009*; *Cifuentes et al., 2010*; *Yamakawa et al., 2014*). Although similar to transcription factors, miRNAs regulate large numbers of target mRNAs and deeply contribute to various cell events, the regulation is mainly required for negative regulation of leaky gene expression and often called fine-tuning (*Sevignani et al., 2006*; *Listowski et al., 2013*). In hematopoiesis, miRNAs are expressed in a lineage-specific manner, and their profiles greatly influence cell differentiation (*Monticelli et al., 2005*; *Neilson et al., 2007*; *Chen et al., 2004*). Focusing on B cell development, it is revealed that Dicer, a key enzyme of miRNA generation, is essential for pre- to pro-B cell transition (*Koralov et al., 2008*). Individual miRNA is also studied, and *Mir150* and *Mir126* are identified as relational factors to B cell lineage development. *Mir150* regulates B cell differentiation by controlling c-Myb expression, and *Mir126* partially rescues *Ebf1*-deficient B cell lineage commitment by modulating IRS-1 expression (*Xiao et al., 2007*; *Okuyama et al., 2013*). Both miRNAs dramatically contributed to B cell development processes, but they were not able to recover B cell development from EBF1 deficiency. Conceived from these vigorous functions of miRNAs on B cell development, at this time, we analyzed ability of *Mir195*, recently revealed as an important factor for several cell differentiation, on B cell lineage commitment in *Ebf1*-deficient HPCs (*Qiu et al., 2017*; *Dueñas et al., 2020*).

## Results

### *Mir195* induces B cell character in *Ebf1*-deficient HPCs

To assess the contribution of *Mir195* on B cell development, Mir195 was transduced into mouse fetal liver (FL)-derived Lin$^-$ c-kit$^+$ HPCs and the cells were differentiated to B220 and CD19 expressing pro-B cells with IL7, Flt ligand, and SCF on OP9 stroma cells. After 7 days of culture, certain numbers of the cells gradually expressed CD19, and the positive cells were increased by *Mir195* transduction (*Figure 1A*). This result suggests that *Mir195* has the ability to shift the HPCs' differentiation toward B cells. Next, we attempted to differentiate *Ebf1*$^{-/-}$ FL HPC to B cell with *Mir195* transduction (*Figure 1B*). As reported in a previous study (*Medina et al., 2004*), control *Ebf1*$^{-/-}$ FL HPCs expressed B220 but did not express CD19. However, *Mir195*-transduced *Ebf1*$^{-/-}$ FL HPCs highly expressed CD19 (*Figure 1C*). In normal B cell development, CD19 expression follows B220 expression, and hence CD19-positive cells show B220 expression as well. Thus, *Mir195*-transduced *Ebf1*$^{-/-}$ FL HPCs, which include B220-negative CD19-positive population, may simply reflect upregulation of CD19 expression, but not B cell development. To exclude this possibility, gene expressions of *Mir195*-transduced *Ebf1*$^{-/-}$ FL HPCs were investigated by cDNA microarray assay and indicated that *Mir195*-transduced cells expressed more B cell lineage-related genes, e.g., *Pax5, Aicda, Rag1, Rag2, Cd79b,* and *Runx2*, whereas less

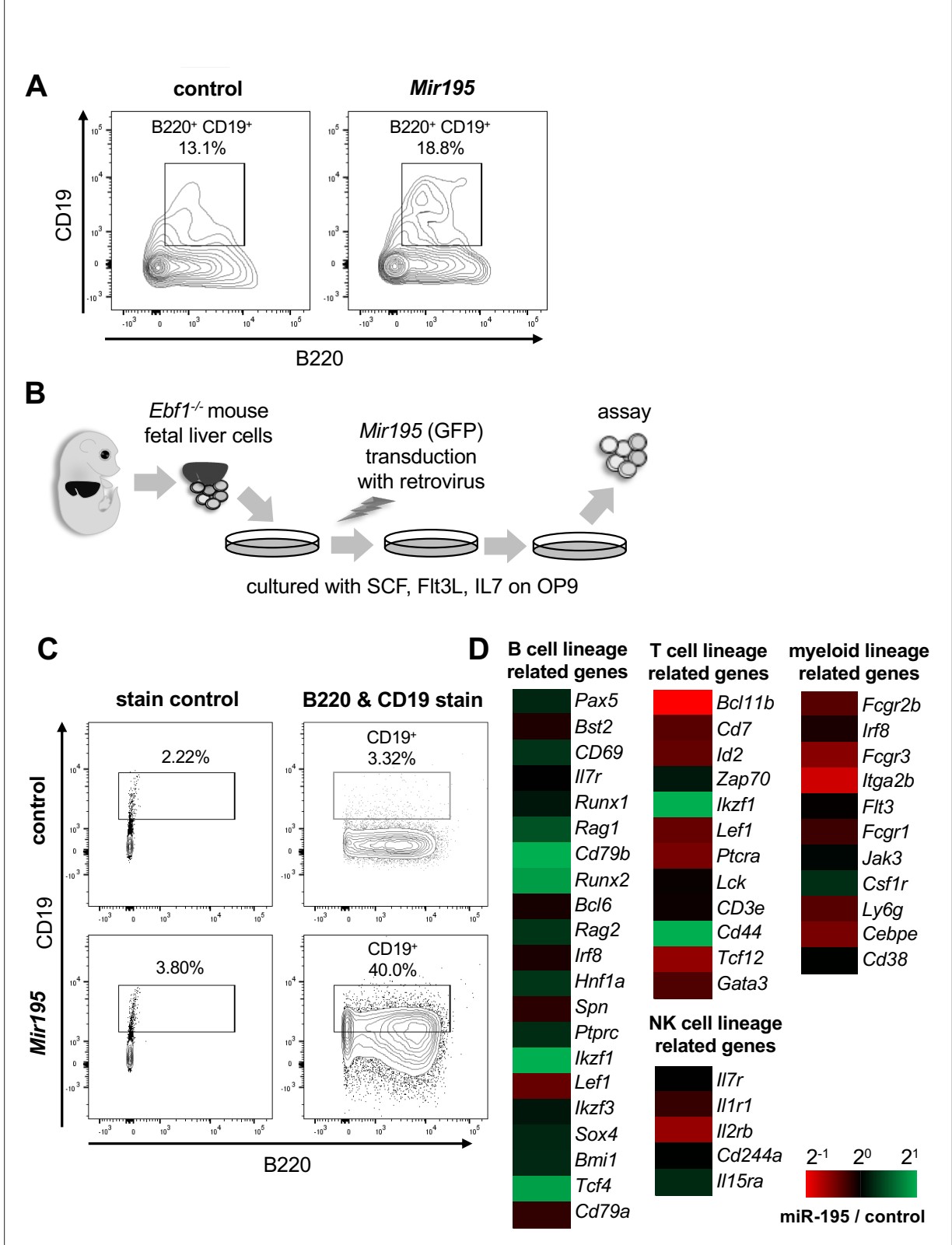

**Figure 1.** *Mir195* promotes hematopoietic progenitor cells (HPCs) to differentiate into the pro-B cell stage without EBF1. (**A**) Flow cytometry analysis of control and *Mir195*-expressing Lin⁻ cells. HPCs from fetal livers of wild-type mice were cultured for 7 days on OP9 with SCF, Flt3-ligand, and IL-7, after infection with control or *Mir195* retrovirus. Representative result of control (upper panel) and *Mir195* (lower panel) viral infections is shown (n=3). (**B**) Outline of the in vitro culture system of *Ebf1⁻/⁻* HPCs. (**C**) Flow cytometry analysis of control and *Mir195*-expressing *Ebf1⁻/⁻* HPCs. Shown data is

*Figure 1 continued on next page*

Figure 1 continued

representative of n=3. (**D**) Microarray analysis of *Mir195*-expressing *Ebf1*[-/-] HPCs. $Log_2$ fold-changes in the expression levels of genes related to B (left panel), T (middle-upper panel), NK (middle-lower panel), and myeloid (right panel) cell lineages were classified and are shown as colored columns. The analysis was carried out in duplicates.

expressed T cell and NK cell lineage-related genes, including *Gata3, Id2*, *Lck*, *Cd3e*, and *Il2rb*, and also myeloid lineage-related genes, e.g., *Cebpe*, *Ly6g*, *Fcgr1*, *Fcgr2b*, and *Fcgr3* (**Figure 1D**). Among the B-lineage transcription factors, *Pax5* and *Erg* were modestly but significantly upregulated ($Log_2$FC ~1.2 and ~0.9, respectively) in *Mir195*-transduced *Ebf1*[-/-] cells compared to controls. While these changes were moderate, they were consistent across replicates and suggest partial restoration of the B cell transcriptional program. These results suggested that not only CD19 expression but also upregulation of several B cell developmental factors and downregulation of other lineage-related genes were involved in the promotion of B cell lineage commitment by *Mir195*.

### *Ebf1*-deficient HPCs were able to commit B cell lineage by transduction of *Mir195* with bone marrow niche modification

The ectopic *Mir195* expression led *Ebf1*[-/-] HPCs to induce differentiation toward B cell. However, a large part of the *Mir195*-transduced HPCs expressed CD19 but not B220, which implied that they strayed from the canonical B cell differentiation steps (**Figure 1C**). In addition to the inner state, the microenvironment known as niche was also critically involved in hematopoiesis (**Cordeiro Gomes et al., 2016**). Especially in early B cell development, bone marrow niches precisely control the maintenance and differentiation of lineage precursors by cytokines and chemokines (**Tokoyoda et al., 2004**). To explore the development of *Mir195*-transduced *Ebf1*[-/-] FL HPCs under bone marrow niches, we engrafted *Mir195*-transduced *Ebf1*[-/-] early B cells into NOG and B6RG mice, in which absence of B cell makes the engrafted B cell visible (**Figure 2A**). After 7 days, the engrafted cells successfully adapted in the bone marrow. While there was no remarkable change in control cell population, notably, instead of B220-negative CD19-positive cells, the double-positive cells were markedly increased in *Mir195*-transduced *Ebf1*[-/-] FL early B cells, suggesting that the normal stepwise B cell development occurred (**Figure 2B**). In B cell development, most prominent steps after CD19 expression are VDJ recombination and subsequent IgM expression on cell surface. In addition to CD19 expression, *Ebf1* is also known as an essential gene for VDJ recombination, especially $V_H$ to $DJ_H$ recombination (**Pongubala et al., 2008**). To determine whether *Mir195*-transduced *Ebf1*[-/-] cells rearranged the VDJ region, we attempted to detect $V_H$-$J_H$ assembled gene segments in the engrafted mouse bone marrow cells by droplet digital PCR (ddPCR). The data revealed that there were a certain number of $V_H$-$J_H$ segments in the bone marrow of mice engrafted with *Mir195*-transduced *Ebf1*[-/-] cells (**Figure 2C**, **Figure 2—figure supplement 1**). Subsequently, to expect the EBF1-independent reconstitution enabled B cell receptor to express as IgM, we analyzed B cell populations in the engrafted mouse bone marrow. Not much, but some *Mir195*-transduced cells expressed IgM on cell surface likely as normal immature B cells in bone marrow 10 days after engraftment (**Figure 2D**). Moreover, these IgM-positive cells were also detected in splenocytes. These data suggested that engrafted cells had differentiated into IgM-positive immature or mature B cells, and they had been recruited to the spleen. The critical function of B cells is changing B cell receptor from IgM to IgG following class switch DNA recombination, which is accompanied by stimuli-induced cell proliferation. To clarify whether *Mir195*-transduced *Ebf1*[-/-] B cells have the function, whole splenocytes of the engrafted mice were stimulated with IL-4 and LPS, which causes class switch recombination to IgG1 (**Muramatsu et al., 1999**). While control GFP-positive cells did not expand by the stimuli, *Mir195*-transduced GFP-positive cells expanded enough to be surely detected, and importantly, a part of them expressed IgG1 (**Figure 2E**). These data suggested that *Mir195* has the potential to induce B cell differentiation from HPCs to mature B cells, resulting in class switch recombination even when critical regulator EBF1 is absent.

### *Mir195* physiologically maintains several B cell populations

As ectopic *Mir195* expression revealed its potential in B cell development. Next, to investigate the contribution of endogenous *Mir195* to B cell lineage populations, *Mir195*-deficient mice in which the genome around *Mir195-5p* was eliminated by CRISPR/Cas9 system were established. The analysis of HPC lineage populations in the bone marrow revealed that several B cell-related progenitors were

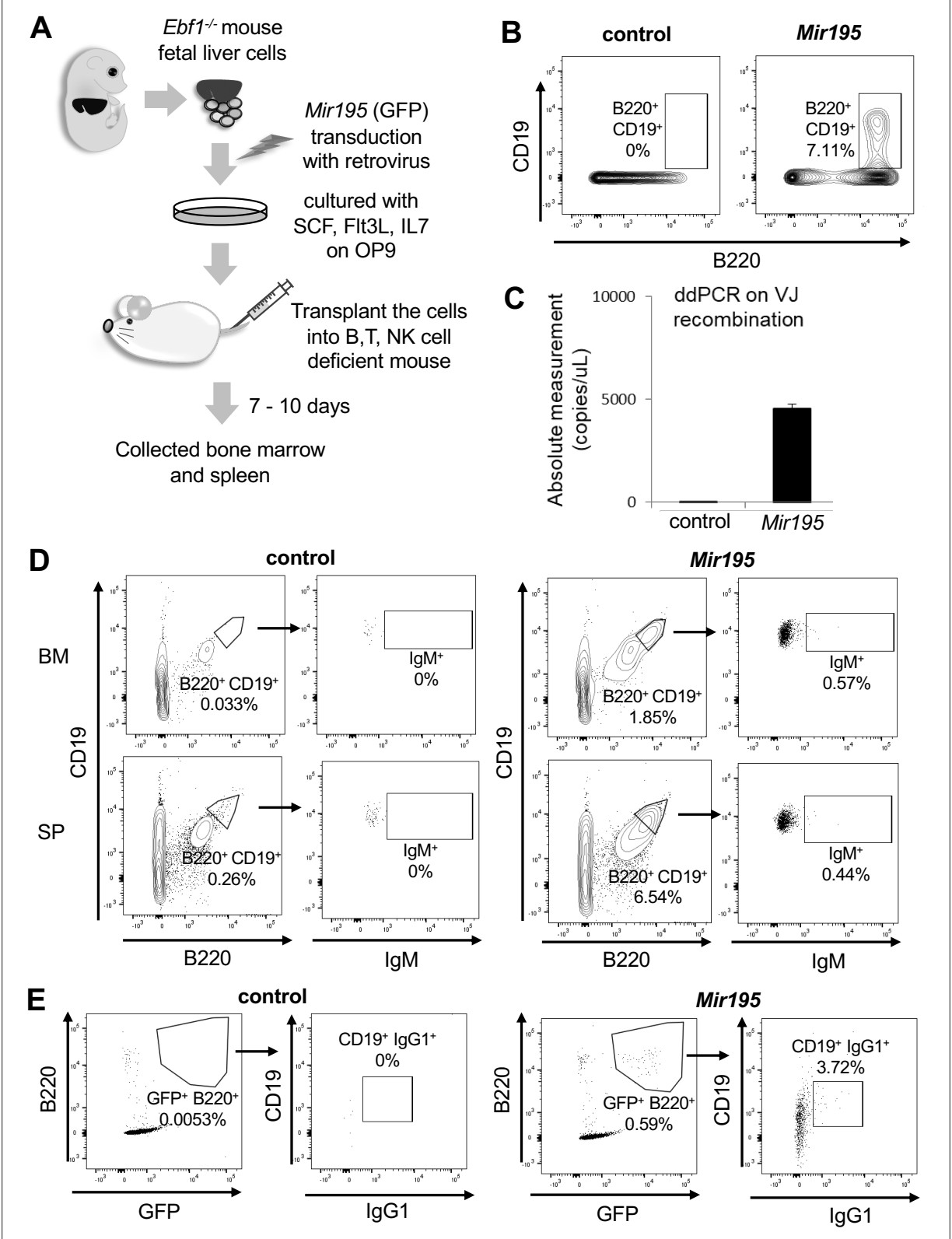

**Figure 2.** *Mir195* leads *Ebf1*-deficient hematopoietic progenitor cells (HPCs) to mature into B cells with bone marrow niche assistance. (**A**) In vivo analysis of B cell development of *Ebf1⁻/⁻* HPCs. (**B**) Flow cytometry analysis of control and *Mir195*-expressing *Ebf1⁻/⁻* HPCs in the bone marrow collected at 7 days after transplantation. (**C**) Using droplet digital PCR, VJ region fragments were amplified from the genomic DNA of B220⁺ cells in the bone marrow of mice transplanted with control and *Mir195*-expressing *Ebf1⁻/⁻* HPCs. (**D**) Flow cytometry analysis of control and *Mir195*-expressing *Ebf1⁻/⁻* HPCs

*Figure 2 continued*

in the bone marrow (BM) and spleen (SP), at 10 days after transplantation. (**E**) Flow cytometry analysis of class-switch recombination. Splenocytes of mice transplanted with control and *Mir195*-expressing *Ebf1*-/- HPCs were cultured for 72 hr with IgG1 class-switch stimuli, LPS, and IL-4. Each flow cytometric data is representative of n=3.

The online version of this article includes the following figure supplement(s) for figure 2:

**Figure supplement 1.** Raw data of *Figure 2C* droplet digital PCR (ddPCR).

relatively reduced in *Mir195*-/- mice. Sca-1- c-kit+ common myeloid progenitor cell population was increased, whereas Sca-1+ c-kit- (LSK-) cells were decreased in *Mir195*-/- mice (**Figure 3A**). As LSK- cells mainly include early lymphoid precursor, these results suggested that *Mir195* is involved in hemato-poiesis, including differentiation of stem cells toward lymphoid and early B cells (**Kumar et al., 2008**). While analysis of each early B cell population did not show significant difference, whole B220+ IgM- pre-B cell populations were slightly increased in the BM of *Mir195*-/- mice (**Figure 3B**). In the splenic B cells, marginal zone B (MZB) cells were reduced in *Mir195*-/- mice (**Figure 3C**). MZB cells were previously reported to be highly dependent on EBF1 activity and disappear in the absence of EBF1. B-1 cells were likewise crucially regulated by EBF1 as well (**Vilagos et al., 2012**). In the peritoneal cavity of *Mir195*-/- mice, B-1 cells were significantly decreased (**Figure 3D**). These results suggested that miR-195 contributed to maintaining several EBF1-dependent mature B cell populations at least in part. Taken together, these results were consistent with those obtained from ectopic expression of *Mir195*.

## FOXO1 phosphorylation pathway targeted by *Mir195* was responsible for B cell lineage commitment

To elucidate how *Mir195* promotes B cell development in *Ebf1*-deficient HPCs, we analyzed regulatory networks of predicted *Mir195* target genes by using starBase_v2.0 and David Bioinformatics Resources 6.8 in KEGG pathway database (**Yang et al., 2011**; **Li et al., 2014**; **Huang et al., 2009a**; **Huang et al., 2009b**; **Kanehisa and Goto, 2000**; **Kanehisa et al., 2016**; **Kanehisa et al., 2017**). Several gene regulation networks were detected as candidates of responsible pathways on the *Mir195* function (**Supplementary file 1 and 2**). Remarkably, MAPK signaling pathway and PI3K-Akt signaling pathway included various targets of *Mir195*. Both MAPK and Akt were known to phosphorylate and degrade FOXO1, which was a critical factor in several stages of B cell development (**Dengler et al., 2008**). Thereby, we focused on the predicted *Mir195* targets: *Pik3r1*, *Pdpk1*, *Akt3*, *Raf1*, *Sos2*, and *Mapk3*, which were involved in and activate MAPK and PI3K-Akt pathways. First, to confirm that the predicted targets are actually regulated by *Mir195*, we picked up the 3'UTR of *Mapk3* and *Akt3*, which were especially important in the pathways, and inserted them in a luciferase reporter assay plasmid. As expected, the luciferase activity was downregulated by *Mir195* transduction, but it was not impaired by transduction of *Mir195* mutant of mature miRNA region (**Figure 4A**). Furthermore, to determine whether the predicted targets were actually regulated by *Mir195*, we measured the expression levels in *Mir195*-transduced *Ebf1*-/- HPCs, and qPCR analysis showed that *Mir195* transduction certainly decreased the mRNA levels (**Figure 4B**). Because of sequence similarity among *Mir15/16* family members, the baseline levels detected in control samples may include signal from endogenous miRNAs such as *Mir497* or *Mir16*. Thus, the observed increase (log$_2$FC~2.5) may underestimate the actual level of *Mir195* overexpression. In line with these findings, *Mapk3* expression was also downregulated in our microarray analysis of *Mir195*-transduced *Ebf1*-/- cells. However, for *Akt3*, the microarray results were inconsistent across different probes, suggesting probe-dependent variability. Therefore, while qPCR and reporter assays support *Akt3* as a potential target of *Mir195*, its regulation remains to be further validated. Next, to evaluate inhibition of FOXO1 phosphorylation and degradation by *Mir195*, we compared protein levels of FOXO1 and phosphorylated FOXO1 (pFOXO1) in *Mir195*-transduced *Ebf1*-/- HPCs. The western blotting results revealed that *Mir195* transduction decreased pFOXO1 levels and increased relative FOXO1 protein levels (**Figure 4C**). We also performed western blotting for PAX5 and ERG using the same samples. The results showed no significant change in these protein levels between *Mir195*-transduced and control *Ebf1*-/- cells (**Figure 4—figure supplement 1**), consistent with the modest upregulation observed in our microarray data. Finally, to determine whether FOXO1 accumulation is sufficient for *Ebf1*-/- HPCs to differentiate into pro-B cells, *Ebf1*-/- HPCs were transduced with *Foxo1* and cultured under the B cell differentiating condition. Similar to

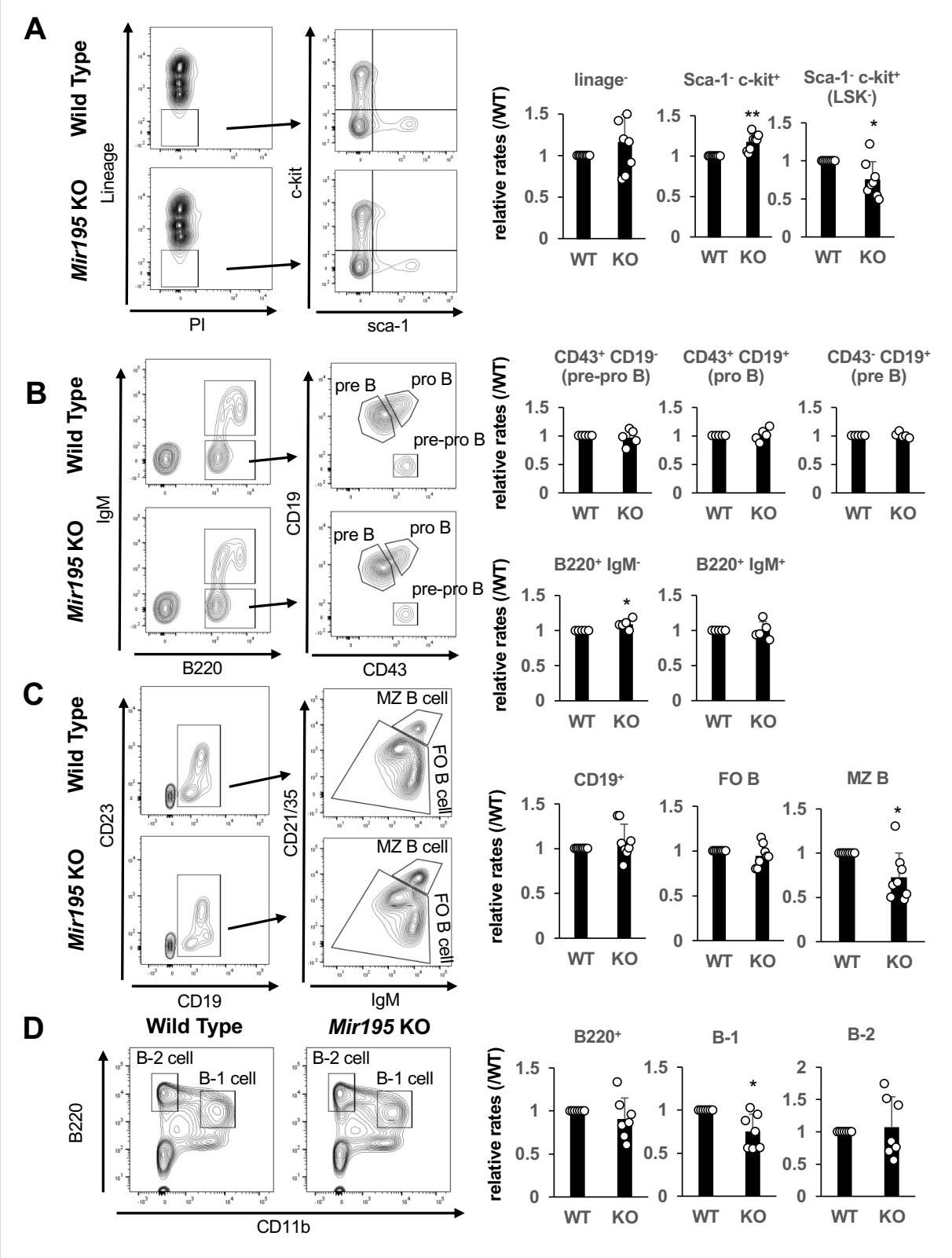

**Figure 3.** Several B cell populations are disturbed in the *Mir195*-deficient mice. Flow cytometry data of B cell lineage populations in *Mir195$^{-/-}$* and littermate WT mice. Representative plots (left side) and mean ± SD of relative population rates in each littermate WT mice (right side) are shown. (**A**) Analysis of early B cell populations in the bone marrow. Pre-pro-B (B220$^+$ IgM$^-$ CD43$^+$ CD19$^-$); pro-B (B220$^+$ IgM$^-$ CD43$^+$ CD19$^+$); pre-B (B220$^+$ IgM$^-$ CD43$^-$ CD19$^+$); n=5. (**B**) Analysis of hematopoietic progenitor populations in the bone marrow; n=5. (**C**) Analysis of B cell populations in the spleen. FO

*Figure 3 continued on next page*

*Figure 3 continued*

B (CD19⁺ IgM⁺ CD21/35^low-middle); MZ B (CD19⁺ IgM⁺ CD21/35^high); n=8. (**D**) Analysis of B cell populations in the peritoneal cavity: B-1 (B220⁺ CD11b⁺); B-2 (B220⁺ CD11b⁻); n=7. Statistical significance was tested using one-sample *t*-test. *p<0.05; **p<0.01. WT, wild-type.

*Mir195* transduction, Foxo1 transduction arose B220 and CD19 double-positive *Ebf1⁻/⁻* cells, which was accompanied by CD19-positive but B220-negative population (*Figure 4D*). These data indicated that FOXO1 accumulation by inhibition of phosphorylating pathways was responsible for *Ebf1⁻/⁻* HPCs to differentiate into B cell lineage.

## Epigenetically activated genes in pro-B cells by *Mir195* are fewer than by EBF1

In B cell development, epigenetic changes of transcription factors and differentiation molecules are crucial for proper development, which are mainly regulated by EBF1 (*Treiber et al., 2010*; *Maier et al., 2004*). We investigated transposase-accessible chromatin using deposited sequencing data (ATAC-seq) of *Ebf1⁻/⁻* pro-B cells and wild-type pro-B cells from GSE92434 and cells in early B cell lineages from GSE100738. While wild-type pro-B cells/*Ebf1⁻/⁻* pro-B cells differentially accessible (DA) ATAC peaks were observed in 2809 sites, wild-type CD19-positive/CD19-negative early B cells DA ATAC peaks were in 904 sites. Then, 678 sites were overlapped, which were considered to be regulated by EBF1 as important locus for early B cell development. Moreover, some of them were overlapped with *Mir195*-transduced B220 and CD19 double-positive *Ebf1⁻/⁻* cells (*Mir195* CD19⁺)/ B220-positive CD19-negative *Ebf1⁻/⁻* cells (control CD19⁻) DA ATAC peaks (73 out of 226 peaks), which were considered to be regulated by *Mir195* (*Figure 5B*). These peaks included important genes for early B cell development, such as *Pax5*, *Runx1*, *Erg*, *Ifr8*, and *Blnk*, and B cell-related genes, such as *Rarres1*, *Ciita*, and *Atg7* (*Supplementary file 3*). These results indicated that gene loci opened by *Mir195* were fewer than by EBF1, but they included several key loci for B cell differentiation, and they were enough to differentiate the progenitor cells to mature B cells. Moreover, HOMER Motif Analysis revealed that enriched motives opened by EBF1 and by *Mir195* were 198 and 111, respectively (*Figure 5C*). The common motifs were 104 which included critical genes for B cell development, such as *E2A*, *Foxo1*, and *Pax5*, and high-ranked motifs were very similar between EBF1 and *Mir195* (*Figure 5D*). These results suggested that *Mir195* transduction opened important chromatin regions for early B cells, which were normally regulated by EBF1. Finally, we concluded that *Mir195* transduction was able to compensate EBF1 deficiency in B cell development through activation of FOXO1 and epigenetic regulation of several B cell-related genes.

## Discussion

The canonical notion of hematopoietic fate determination implies that EBF1 is an indispensable factor for B lymphopoiesis. However, in this study, we showed that a single miRNA *Mir195* rescued the arrest of pro-B cell differentiation induced by EBF1 deficiency. As miRNA plays roles in a bundle of their family, single miRNA-deficient mice often do not show significant phenotype (*Song et al., 2014*). Nevertheless, *Mir195*-deficient mice showed a small but consistent decrease in the number of several hematopoietic cells, including MZB cells and peritoneal B-1 cells, which were reported to almost disappear in *Ebf1^ihCd2* mice in which EBF1 was deficient in mature B cells (*Vilagos et al., 2012*). Considering that other miRNA-deficient mice have subtle phenotypes and *Mir195* is one of the large family, including *Mir15/16* and *Mir195/497* (*Hutter et al., 2021*), the remarkable potential of *Mir195* is beyond a fine tuner as miRNA, at least as far as it is considered with regard to B cell lineage commitment.

A part of the mechanisms of the potent function of *Mir195* was caused by inhibition of phosphorylation of FOXO1. FOXO1 is a transcriptional factor controlled by EBF1 and strongly promotes differentiation of pre-B cells. FOXO1 activity is regulated by the PI3K/AKT pathway, and several miRNAs were reported to be involved in the regulation (*Coffre et al., 2016*). We showed *Foxo1* transduction enabled *Ebf1*-deficient cells to express CD19. However, the CD19-positive cells rapidly disappeared and couldn't be detected in transplanted mice (data not shown). It is presumable that FOXO1 activity was necessary to express CD19, but other factors undertake maintenance and proliferation of the developing cells. ATAC-seq analysis revealed that *Mir195* was directly or indirectly involved in

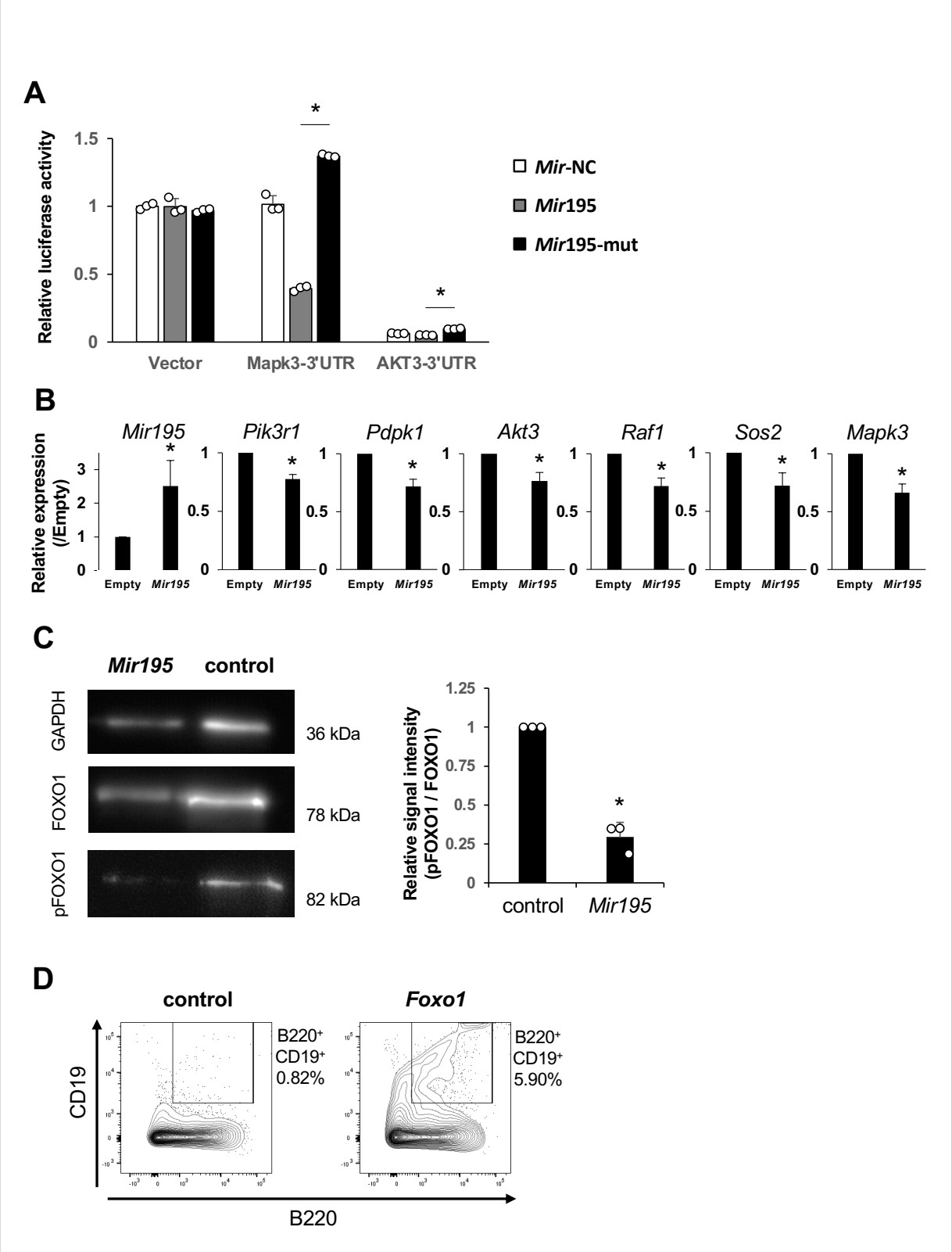

**Figure 4.** FOXO1 phosphorylation pathways are key targets of *Mir195* for promotion of B cell development. (**A**) Relative luciferase inhibitory rates of *Mir195* onto predicted target 3′UTR were analyzed using Dual-Luciferase reporter assay. (**B**) Relative expression rate of *Mir195* and predicted target genes were compared between control (EMPTY) and *Mir195*-expressing *Ebf1*[-/-] hematopoietic progenitor cells (HPCs). (**C**) Western blot of FOXO1 and phosphorylated FOXO1 (pFOXO1) in control and *Mir195*-expressing *Ebf1*[-/-] HPCs. Quantification of FOXO1 and phospho-FOXO1 band intensities from

*Figure 4 continued on next page*

*Figure 4 continued*

three independent experiments is shown in the bar graph. Data are presented as mean ± SD. Shown data is representative of n=3. (**D**) Flow cytometry analysis of control and *Foxo1*-expressing *Ebf1*⁻/⁻ HPCs. Shown data is representative of n=3. Statistical significance was tested using a one-sample *t*-test. *p<0.05, n=3.

The online version of this article includes the following source data and figure supplement(s) for figure 4:

**Source data 1.** PDF file containing original western blots for *Figure 4C*, indicating the relevant bands and treatments.

**Source data 2.** Original files for western blot analysis displayed in *Figure 4C*.

**Figure supplement 1.** Western blot analysis of other known B-lineage regulators in Ebf1⁻/⁻ hematopoietic progenitor cells (HPCs) with or without Mir195 transduction.

**Figure supplement 1—source data 1.** PDF file containing original western blots for *Figure 4—figure supplement 1*, indicating the relevant bands and treatments.

**Figure supplement 1—source data 2.** Original files for western blot analysis displayed in *Figure 4—figure supplement 1*.

chromatin accessibility. As the chromatin regions and motifs opened by *Mir195* were critical for B cell differentiation and hematopoiesis, further investigation is needed for the mechanism.

Although our study indicates that *Mir195* has the potential to promote B cell lineage commitment in the absence of EBF1, the precise downstream targets and mechanisms remain only partially defined. We hypothesize that the observed effects are mediated through the downregulation of multiple mRNA targets involved in opposing B-lineage differentiation, including kinases in the MAPK and PI3K-Akt pathways that modulate FOXO1 phosphorylation. While our microarray, qPCR, and luciferase assays support the regulation of specific targets such as *Mapk3* and *Akt3*, a more comprehensive identification of direct targets—especially those related to transcriptional and epigenetic regulation—would further strengthen our conclusions. We interpret our findings as revealing the potential of *Mir195* to compensate for EBF1 deficiency, rather than a demonstration of its physiological role. Future studies using global transcriptome, proteome, and chromatin-binding assays will be essential to fully elucidate the mechanisms underlying this observation.

To compensate for the lack of transcriptome data from sorted *Mir195*-transduced pre-pro-B or CD19⁺ *Ebf1*⁻/⁻ cells, we compared our microarray data with publicly available RNA-seq profiles of *Ebf1*⁻/⁻ pro-B cells (GSE92434). This analysis revealed that several B-lineage defining genes downregulated in EBF1 deficiency were upregulated upon *Mir195* expression, suggesting that *Mir195* may partially restore transcriptional programs disrupted by the loss of EBF1.

Although direct evidence of FOXO1 binding to B-lineage gene loci (e.g. via ChIP-seq or CUT&RUN) is currently lacking due to technical limitations in cell numbers, our results suggest that FOXO1 plays a key functional role. This is supported by its increased protein level upon *Mir195* expression, the partial phenocopy by FOXO1 overexpression, and the enrichment of FOXO1 motifs in open chromatin regions identified by ATAC-seq. Future studies incorporating FOXO1 chromatin profiling will be important to validate its direct regulatory role in this context.

While ddPCR provided a sensitive means to detect $V_H$-$J_H$ rearranged fragments, it does not offer resolution of specific V, D, or J gene usage or recombination completeness. Therefore, the full extent and diversity of V(D)J recombination in *Ebf1*⁻/⁻ *Mir195*-transduced CD19⁺ cells remains to be clarified. Future studies incorporating high-throughput sequencing approaches will be important to fully characterize the immunoglobulin repertoire and confirm progression through the pre-BCR checkpoint.

While our data support the B-lineage identity of *Mir195*-transduced *Ebf1*⁻/⁻ CD19⁺ cells based on gene expression, chromatin accessibility, and immunoglobulin expression, we have not directly tested their lineage plasticity under alternative differentiation conditions. Whether these cells retain responsiveness to myeloid cytokines or exhibit residual multipotency remains to be determined. Future studies using single-cell fate mapping or in vitro differentiation assays will be required to fully define the lineage commitment status of this population.

While our results demonstrate that ectopic expression of *Mir195* can compensate for the loss of EBF1 in promoting B cell development, we acknowledge that this does not necessarily reflect a physiological role for *Mir195*. The *Mir195* knockout mice exhibited only mild alterations in B cell populations, suggesting that under normal conditions, *Mir195* is not essential for B lymphopoiesis. Therefore, our findings should be interpreted as highlighting the potential of *Mir195* to modulate B cell fate under specific conditions, rather than indicating its requirement in physiological B cell

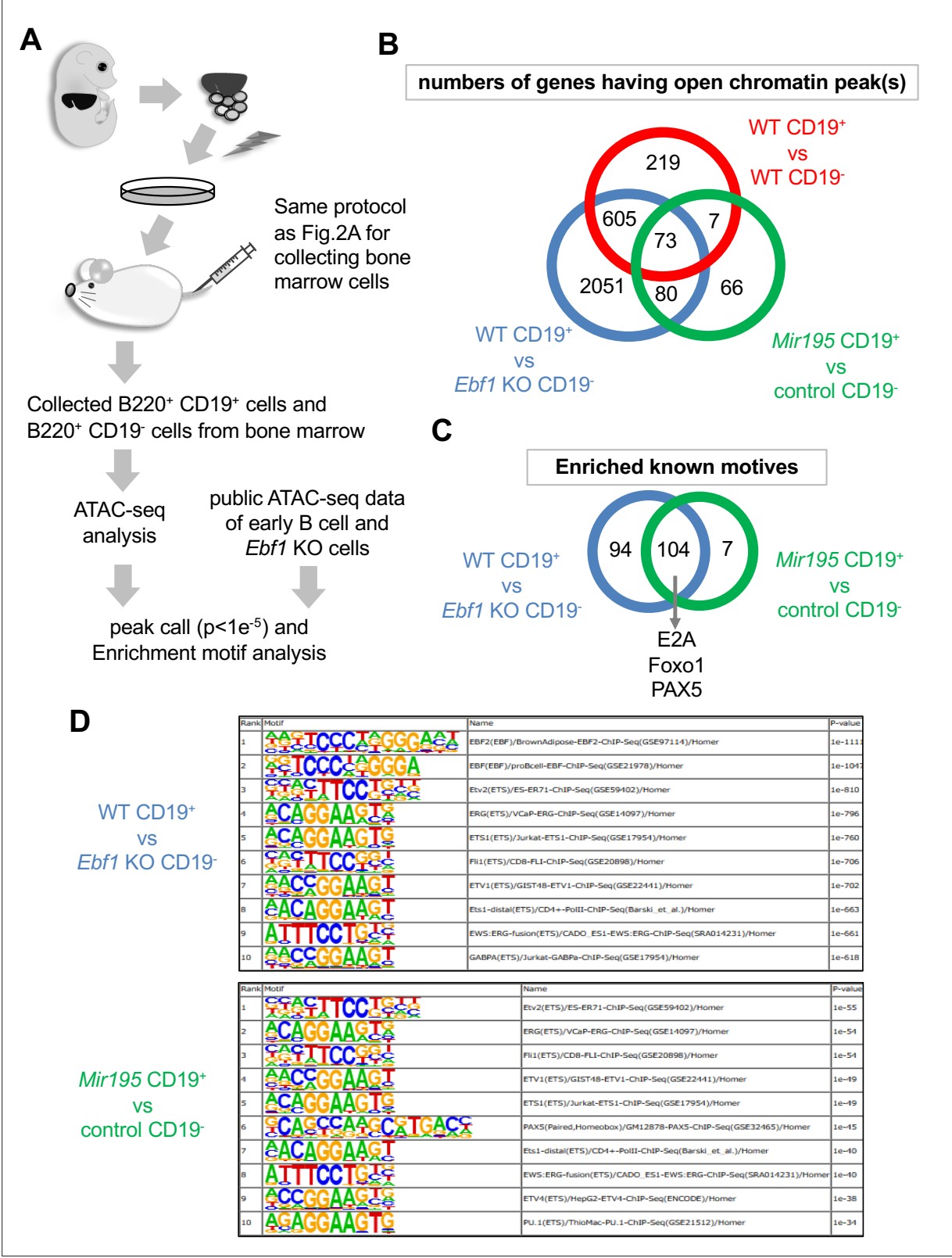

**Figure 5.** ATAC-seq analysis of *Ebf1*[-/-] CD19-positive B cells differentiated by *Mir195*. (**A**) Outline of analysis of open chromatin regions in *Mir195*-expressing *Ebf1*[-/-] cells. (**B**) Venn diagram of numbers of genes in which DNA regions of open chromatin peaks were detected by means of peak call analysis. The analyses were examined between CD19-negative (FrA) and -positive (FrB, FrC, and FrD) stages of B cell development (GSE100738; upper red circle); wild-type (WT) and *Ebf1*[-/-] pro-B cells (GSE92434; left-lower blue circle); B220+ CD19- cells of control and B220+ CD19+-positive miR-195-

*Figure 5 continued*

expressing *Ebf1^-/-* cells (right-lower green circle). Overlapping regions in the Venn diagram are interpreted as follows: the intersection of WT and Rescue represents canonical EBF1-regulated regions; the overlap between Rescue and *Mir195* indicates partial mimicry by *Mir195*; and regions unique to *Mir195* may reflect EBF1-independent chromatin changes. (**C and D**) Venn diagram of numbers of enriched known motifs detected using HOMER find motif analysis (**C**) and lists of high p-value motifs, up to rank 10 (**D**).

development. Further studies will be needed to determine whether *Mir195* plays a more prominent role under stress or disease contexts, or in cooperation with other miRNAs.

The luciferase activity was markedly reduced in the presence of the Akt3 3'UTR, even in cells transduced with a control vector (**Figure 4A**). We hypothesize that the Akt3 3'UTR contains strong posttranscriptional regulatory elements—such as AU-rich elements or binding sites for endogenous miRNAs or RNA-binding proteins—which may suppress mRNA stability or translation independent of *Mir195*. Alternatively, the secondary structure or length of the UTR may inherently reduce luciferase expression.

## Materials and methods

### Plasmid construction

To construct MDH1-PGK-GFP-Mir195, genomic DNA was first extracted from RS4;11 using the DNeasy Tissue Extraction Kit (QIAGEN). Next, a segment around *Mir195* was amplified by means of PCR, using Pfx polymerase (Invitrogen) and the oligonucleotides, 5′-AGATCTCTCGAGAAGGAGAG GGTGGGGTAT-3′ and 5′-GGGGCGGAATTCGCTATTCCCGCATAAGCA-3′. The obtained PCR product was then cloned into the XhoI-EcoRI site of MDH1-PGK-GFP 2.0 (Addgene #11375). To construct pMYs-RFP-Foxo1, first, pEX-Foxo1 (in which mouse Foxo1 is optimized for gene synthesis; Eurofins Genomics KK) was synthesized and inserted into the EcoRI-XhoI site of pEX. Next, the Foxo1 region was extracted using the restriction enzymes and inserted into pMYs-RFP retroviral vector (kindly provided by Prof. T Kitamura, Tokyo University). For in vitro transcription of small-guide RNA (sgRNA), pUC57-195sg-upstream and -downstream were generated. Both plasmids originated from the pUC57-sgRNA expression vector (Addgene #51132), and the annealed oligonucleotides were inserted into a BsaI site. (For the former, 5′-TAGGCCCACAAAGGCAGGGACCTA-3′ and 5′-AAACTAGGTCCCTGCC TTTGTGGG-3′ were annealed, while for the latter, 5′-TAGGGGAAGTGAGTCTGCCAATAT-3′ and 5′-AAACATATTGGCAGACTCACTTCC-3′ were annealed.) For the Dual-Luciferase assay, psiCHECK-2 vector was purchased from Promega and the 3′-UTRs of Akt3 and Mapk3 were inserted between the XhoI and NotI sites. MDH1-PGK-GFP-Mir195-mut was generated by mutating 6 bases, from the second to seventh bases of the mature Mir195 and complementary regions of the stem loop structure in MDH1-PGK-GFP-Mir195. In detail, normal Mir195 stem loop sequence 5′-AGCUUCCCUGGC UCUAGCAGCACAGAAAUAUUGGCACAGGGAAGCGAGUCUGCCAAUAUUGGCUGUGCUGCU CCAGGCAGGGUGGUG-3′ (mature Mir195-5p sequence 5′-UAGCAGCACAGAAAUAUUGGC-3′) was mutated to 5′-AGCUUCCCUGGCUCUgcgccgACAGAAAUAUUGGCACAGGGAAGCGAGUCUG CCAAUAUUGGCUGUcggcgcCCAGGCAGGGUGGUG-3′ (mature sequence 5′-UgcgccgACAGA AAUAUUGGC-3′).

### Animals

C57BL/6 mice were purchased from CLEA Japan Inc NOD/Shi-scid, IL-2RγKO (NOG) and B6RG mice were purchased from Central Institute for Experimental Animals (CIEA). The *Ebf1^+/-* mice were originally generated by R Grosschedl (**Lin and Grosschedl, 1995**). *Mir195*-deficient mice were generated based on the CRISPR/Cas9 system established by C Gurumurthy (**Harms et al., 2014**), using pUC57-195sg-upstream and -downstream for sgRNA expression and pBGK (Addgene #65796) for Cas9 mRNA expression. Sanger sequencing confirmed a deletion of 5103 base pairs at chromosome 11 (GRCm38/mm10 chr11:70,234,425–70,235,103), encompassing the entire *Mir497* sequence upstream and 61 bp of the 93 bp *Mir195* precursor. The deletion was validated using genomic DNA and aligned to the mouse reference genome. All transgenic mice used for experiments were backcrossed to the C57BL/6 background for at least eight generations to minimize off-target effects. The obtained mice were subsequently bred and housed at Tokai University. All the animal experiments in this study were approved by the Institutional Review Board of Tokai University (211039, 221046, 231073, 231116,

241089). All the animal experiments in this study complied with the Guidelines for the Care and Use of Animals for Scientific Purposes at Tokai University. To reduce the number of sacrificed animals, the sample sizes for each animal experiment were empirically determined from previous studies or the results of the first littermate mice.

## Flow cytometry analysis

Cells were collected and washed in FACS buffer (phosphate-buffered saline supplemented with 2% fetal bovine serum) and subsequently stained with the following antibodies purchased from BioLegend: anti-c-kit (2B8), -Sca-1 (D7), -IL7Rα (A7R34), -B220 (RA3-6B2), -IgM (RMM-1), -CD3ε (145-2 C11), -CD4 (GK1.5), -CD8 (53–6.7), -CD11b (M1/70), -CD19 (1D3), -CD23 (B3B4), and -IgG1 (RMG1-1) and Thermo Fisher: anti-Flt3 (A2F10), -CD43 (eBioR2/60), and -CD21/35 (eBio8D9). All samples were analyzed on the BD FACSVerse system, and the data obtained was analyzed using FlowJo. FACSAria III was used for cell sorting.

## Culture of lineage-negative (Lin⁻) cells from the FL

FLs were harvested from pregnant C57BL/6 or *Ebf1*$^{+/-}$ (mated with *Ebf1*$^{+/-}$ male) at 13.5 days after vaginal plug formation and minced gently by means of pipetting. The cell suspensions were filtered through a 67 µm pore nylon mesh and Lin⁻ cells were collected using the Lineage Cell Depletion Kit, mouse and AutoMACS Pro Separator (Miltenyi Biotec), according to the manufacturer's instructions. Subsequently, the collected Lin⁻ cells were transduced with *Mir195* or *Foxo1* by means of retroviral transfection. In brief, Platinum-E cells were transfected with MDH1-PGK-GFP (for EMPTY sample) or MDH1-PGK-GFP-Mir195 or pMYs-RFP-Foxo1 using PEI MAX (Polysciences Inc), and retroviral super-natants were harvested 48 hr later. Lin⁻ cells were infected with the supernatants using 10 µg/mL Polybrene (Sigma-Aldrich). The infected and transduced Lin⁻ cells were cultured and differentiated into B cells on OP9 cells in IMDM (Thermo Fisher) supplemented with 10% fetal bovine serum, 1 mM sodium pyruvate, 0.1 mM non-essential amino acid solution, 50 µM 2-mercaptoethanol, 100 units/mL penicillin G, 100 µg/mL streptomycin (all from Wako), and 10 ng/mL recombinant SCF, IL-7, and Flt3-ligand (R&D Systems). Cells were cultured on OP9 cells for 7 days before analysis unless otherwise specified. For in vivo analysis of B cell development of *Ebf1*$^{-/-}$ Lin⁻ cells, 1×10$^6$ cells were injected into the NOG or B6RG mice after >7 days of culture and expansion in vitro.

## Microarray analysis

Total RNA was isolated using the RNeasy MINI Kit (QIAGEN), and its quality was analyzed using the 2100 Bioanalyzer (Agilent Technologies). Approximately 100 ng RNA was labeled, and gene expression microarray analysis was performed using the Agilent Whole Mouse Genome Microarray 4×44K v2 (Agilent Technologies), according to the manufacturer's instructions. The processed data was analyzed using GeneSpring GX version 14.9 (Agilent Technologies). Raw intensity values were normalized using the 75th percentile and transformed to the Log$_2$ scale. All experiments were carried out in duplicates.

## Droplet digital PCR

To carry out ddPCR for VJ recombination analysis, total DNA was isolated from whole cells of the bone marrow in *Mir195*-transduced *Ebf1*$^{-/-}$ FL HPCs-engrafted NOG mice, using the Wizard Genomic DNA Purification Kit (Promega). ddPCR was conducted using QX100 Droplet Digital PCR system (Bio-Rad). Briefly, 3.3 µL of template cDNA with 20× primer and a TaqMan probe set was partitioned into approximately 20,000 droplets using the QX100 Droplet Generator, for amplification. The cycling conditions were 95°C for 10 min, followed by 50 cycles of 95°C for 15 s and 60°C for 1 min, and a final 10 min incubation at 98°C. The droplets were subsequently read automatically using the QX10 droplet reader. The data were analyzed with QuantaSoft analysis software (ver. 1.3.2.0; Bio-Rad). The primers used were as follows: forward primer – 5'-GAGGACTCTGCRGTCTATTWCTGTGC-3'; reverse primer – 5'- CCCTGACCCAGACCCATGT-3'; and probe – 5'-6FAM-TTCAACCCCTTTGTCCCAAAGTT-TAM-3'.

## Class-switch stimulation

*Ebf1*$^{-/-}$ Lin⁻ cells were transduced with EMPTY and *Mir195*-expressing vector and transplanted into B6RG mice. At 10 days posttransplantation, the spleens were collected from the mice, minced with slide glasses, and filtered through a 67 µm pore nylon mesh. IgM⁺ cells were sorted and stimulated

for 3 days with 12.5 µg/mL lipopolysaccharide (Sigma-Aldrich) and 7.5 ng/mL IL-4 (Peprotech) in RPMI-1640 (Wako) supplemented with 10% fetal bovine serum, 100 U/mL penicillin G, and 100 µg/mL streptomycin.

## Gene Ontology analysis

The *Mir195* targetomes were gathered from the *Mir195* target mRNAs identified from three databases (TargetScan, miRDB, and microRNA.org) and by comparing the microarray data of the targets in control- and *Mir195*-transduced *Ebf1*$^{-/-}$ FL HPCs. To investigate the biological functions, these genes were applied to the Gene Ontology classification using GeneSpringGX11.

## Quantitative real-time PCR

For mRNA quantification, total RNA was isolated using Sepasol-RNA I Super G (Nacalai Tesque), and cDNA was synthesized from it using the ReverTra Ace qPCR RT Master Mix (TOYOBO). qPCR was performed using THUNDERBIRD SYBR qPCR Mix (TOYOBO) on the StepOnePlus Real-Time PCR System (Thermo Fisher). The following primers were used for qPCR: Pik3r1 – 5′-AAACTCCGAGAC ACTGCTGA-3′ and 5′-GAGTGTAATCGCCGTGCATT-3′; Pdpk1 – 5′-CTGGGCTCTGCTCTAGTGTT-3′ and 5′-CCCAGGTTCAGGACAGGATT-3′; Akt3 – 5′-GTGGACCACTGTTATAGAGAGAACAT-3′ and 5′-TTGGATAGCTTCCGTCCACT-3′; Raf1 – 5′-TCTTCCATCGAGCTGCTTCA-3′ and 5′-GGATGTAGTCAG CGTGCAAG-3′; Sos2 – 5′-AACTTTGAAGAACGGGTGGC-3′ and 5′-TTTCCTGCAGTGCCTCAAAC -3′; and Mapk3 – 5′-ACTACCTGGACCAGCTCAAC-3′ and 5′-TAGGAAAGAGCTTGGCCCAA-3′. For *Mir195* quantification, TaqMan MicroRNA Assay (ABI) was used. Briefly, total RNA was isolated using Sepasol-RNA I Super G, and cDNA was synthesized from it using the microRNA TaqMan MicroRNA Reverse Transcription Kit (Thermo Fisher) and a specific primer, 5′-UAGCAGCACAGAAAUAUUGGC-3′. The expression levels were measured using the TaqMan Fast Advanced Master Mix (Thermo Fisher) on the StepOnePlus Real-Time PCR System. Given the high sequence similarity among *Mir15/16* family members, the TaqMan assay for *Mir195* may detect related miRNAs such as *Mir16*. Therefore, we interpreted *Mir195* qPCR results as approximate estimates rather than precise quantification. GAPDH was used for normalization to maintain consistency with other qPCR assays in this study. All reagents and kits in this section were used according to the manufacturer's instructions. Target RNA expression levels were compared with those of GAPDH using the $2^{-\Delta\Delta Ct}$ method.

## Dual-Luciferase assay

293T cells were co-transfected with 20 ng psiCHECK-2 of *Akt3* or *Mapk3* and 100 ng MDH1-PGK-GFP-Mir195 or MDH1-PGK-GFP-Mir195-mut. At 48 hr post-transfection, the relative amounts of Renilla and firefly luciferase were analyzed using a Dual-Luciferase Reporter Assay System (Promega). The Renilla/firefly luciferase ratio was calculated and normalized against the control.

## Western blot

Total proteins were collected from whole cells using radioimmunoprecipitation assay buffer (Wako) with protease inhibitor cocktail (Sigma-Aldrich) and SDS sample buffer (60 mM Tris-HCl pH 6.8, 2% SDS, 10% glycerol, and 50 mM dithiothreitol). The proteins were separated using SDS-PAGE, and the western blot signal was detected and analyzed using the Immobilon Western Chemiluminescent HRP Substrate (Millipore) on Ez-Capture MG AE-9300 (ATTO). The following antibodies were used: anti-FOXO1 (C29H4, Cell Signaling Technology), anti-phospho-FOXO1(Ser256) (9461, Cell Signaling Technology), and anti-GAPDH (G9545, Sigma-Aldrich), anti-PAX5 (26709-1-AP, Proteintech), anti-ERG (14356-1-AP, Proteintech), anti-β-actin (A5441, Sigma-Aldrich). Signal intensities were quantified using ImageJ version 1.54g (*Schneider et al., 2012*).

## ATAC-seq analysis

For ATAC-seq analysis, B220$^+$ CD19$^+$ and B220$^+$ cells were sorted from the bone marrow of NOG mice transplanted with *Mir195*-transduced *Ebf1*$^{-/-}$ Lin$^-$ cells. B220$^+$ cells were also sorted from the empty transduced sample. The collected cells were resolved using CELLBANKER (Takara Bio) and temporarily preserved at –20°C. ATAC-seq libraries were prepared from the cryopreserved cells according to the Omni-ATAC protocol (*Corces et al., 2017*). Briefly, >5000 cells were lysed and subjected to a transposition reaction. The transposed fragments were pre-amplified, quantitated using RT-PCR, and

then amplified again. The prepared libraries were sequenced on the NextSeq 550 platform (Illumina) with paired-end reads (read 1, 75 bp; index 1, 8 bp; index 2, 8 bp; read 2, 75 bp). Short-read data were trimmed using sickle 1.33 (https://github.com/najoshi/sickle; *Joshi and Fass, 2011*) and mapped onto a mm10 reference genome using bowtie2. Unmapped, multi, chrM mapping, and duplicate reads were eliminated using samtools 1.16.1 and Picard Tools (Picard MarkDuplicates; http://broadinstitute.github.io/picard). Peak summits in all populations were determined using the MACS3 functions (-callpeak -p 1e-5; https://github.com/macs3-project/MACS; *Lui et al., 2025*). Motif enrichment analysis was carried out using HOMER, with default settings.

## Statistical analysis

One-sample *t*-test was used to analyze differences between groups, and p-values<0.05 were considered statistically significant. All analyses were performed using Excel (Microsoft). Statistical significance was determined using the Fisher's exact test, followed by multiple test corrections using the Benjamini and Yekutieli false discovery rate method.

## Acknowledgements

We thank N Kurosaki, K Takahashi, E Nagashima, and members of the Department of Innovative Medical Science at Tokai University for their assistance, advice, and helpful discussions. We also thank the Support Center for Medical Research and Education at Tokai University for their technical assistance. This work was supported by Grants-in-Aid for Scientific Research JP20H03716 (to AK) and JP20K17362 (to YM) from the Japan Society for the Promotion of Science; P-PROMOTE 22ama221213 and 22ama221215 (to AK) from the Japan Agency for Medical Research and Development; and JST-CREST JPMJCR19H5 (to AK) from the Japan Science and Technology Agency.

## Additional information

### Competing interests

Tomohiro Kurosaki: Reviewing editor, *eLife*. The other authors declare that no competing interests exist.

### Funding

| Funder | Grant reference number | Author |
| --- | --- | --- |
| Japan Society for the Promotion of Science | JP20H03716 | Ai Kotani |
| Japan Society for the Promotion of Science | JP20K17362 | Yuji Miyatake |
| Japan Agency for Medical Research and Development | 22ama221213 | Ai Kotani |
| Japan Agency for Medical Research and Development | 22ama221215 | Ai Kotani |
| Japan Science and Technology Agency | JPMJCR19H5 | Ai Kotani |

The funders had no role in study design, data collection and interpretation, or the decision to submit the work for publication.

### Author contributions

Yuji Miyatake, Data curation, Formal analysis, Funding acquisition, Validation, Investigation, Visualization, Methodology, Writing – original draft, Project administration, Writing – review and editing; Takeshi Kamakura, Data curation, Formal analysis, Investigation, Writing – original draft, Writing – review and editing; Tomokatsu Ikawa, Ken-ichi Hirano, Hiroyuki Hosokawa, Katsuto Hozumi, Masato Ohtsuka, Takahiro Kisikawa, Chikako Shibata, Motoyuki Otsuka, Kiyoshi Ando, Tomohiro

Kurosaki, Resources; Ryo Yanagiya, Ryutaro Kotaki, Kazuaki Kameda, Ryo Koyama Nasu, Kazuki Okuyama, Data curation; Reo Maruyama, Resources, Software; Hiroshi Kawamoto, Formal analysis, Validation; Ai Kotani, Conceptualization, Supervision, Funding acquisition, Validation, Investigation, Visualization, Methodology, Writing – original draft, Project administration, Writing – review and editing

### Author ORCIDs
Takeshi Kamakura (ID) https://orcid.org/0009-0002-4523-8971
Kazuaki Kameda (ID) https://orcid.org/0000-0001-8418-5118
Ken-ichi Hirano (ID) https://orcid.org/0000-0003-3495-610X
Hiroyuki Hosokawa (ID) https://orcid.org/0000-0002-9592-2889
Katsuto Hozumi (ID) https://orcid.org/0000-0002-7685-6927
Tomohiro Kurosaki (ID) https://orcid.org/0000-0002-6352-304X
Ai Kotani (ID) https://orcid.org/0000-0002-0976-8687

### Ethics
All the animal experiments in this study were approved by the Institutional Review Board of Tokai University (211039, 221046, 231073, 231116, 241089). All the animal experiments in this study complied with the Guidelines for the Care and Use of Animals for Scientific Purposes at Tokai University. To reduce the number of sacrificed animals, the sample sizes for each animal experiment were empirically determined from previous studies or the results of the first littermate mice.

Reviewer #1 (Public review): https://doi.org/10.7554/eLife.101510.3.sa1
Reviewer #2 (Public review): https://doi.org/10.7554/eLife.101510.3.sa2
Reviewer #3 (Public review): https://doi.org/10.7554/eLife.101510.3.sa3
Author response https://doi.org/10.7554/eLife.101510.3.sa4

## Additional files

### Supplementary files
MDAR checklist

Supplementary file 1. Lists of high-p-value pathways detected by means of targetome analysis of human Mir195 with TargetScan and starBase in the KEGG pathway.

Supplementary file 2. Lists of high-p-value pathways detected by means of targetome analysis of murine Mir195 with TargetScan and starBase in the KEGG pathway.

Supplementary file 3. List of genes with detected peaks in the DNA region common in the three analyses.

Supplementary file 4. Targetome analysis of human Mir195 with TargetScan.

Supplementary file 5. Targetome analysis of murine Mir195 with TargetScan.

### Data availability
The microarray data was deposited in Gene Expression Omnibus with the identifier GSE246669, and the ATAC-seq data was also deposited with the identifier GSE246530. The other data generated in this study are available in the manuscript or supplementary materials.

The following datasets were generated:

| Author(s) | Year | Dataset title | Dataset URL | Database and Identifier |
|---|---|---|---|---|
| Miyatake Y, Yanagiya R, Kotani A | 2024 | Analysis of expression profiles in miR-195 transduced Ebf1-/- Lin- cells | https://www.ncbi.nlm.nih.gov/geo/query/acc.cgi?acc=GSE246669 | NCBI Gene Expression Omnibus, GSE246669 |
| Miyatake Y, Yanagiya R, Maruyama R, Kotani A | 2024 | ATAC-seq data of miR-195 transduced Ebf1-KO FL-ProB-cells | https://www.ncbi.nlm.nih.gov/geo/query/acc.cgi?acc=GSE246530 | NCBI Gene Expression Omnibus, GSE246530 |

The following previously published datasets were used:

| Author(s) | Year | Dataset title | Dataset URL | Database and Identifier |
|---|---|---|---|---|
| Yoshida H, Ramirez R, Lareau C, Buenrostro J, Rhoads A, Mostafavi S, Mathis D, Benoist C | 2017 | ImmGen ATAC-seq data | https://www.ncbi.nlm.nih.gov/geo/query/acc.cgi?acc=GSE100738 | NCBI Gene Expression Omnibus, GSE100738 |
| Okuyama K, Strid T, Somasundaram R, Prasad M, Åhsberg J, Sigvardsson M | 2018 | RNA- and ATAC-seq data of Wt, Ebf1-KO and Pax5-KO FL-ProB-cells | https://www.ncbi.nlm.nih.gov/geo/query/acc.cgi?acc=GSE92434 | NCBI Gene Expression Omnibus, GSE92434 |

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
