## [Editor Report · eLife Assessment]

This **useful** study reports that the exogenous expression of the microRNA miR-195 can partially compensate in early B cell development for the loss of EBF1, one of the key transcription factors in B cells. While this finding will be of interest to those studying lymphocyte development, the evidence, particularly with regard to the molecular mechanisms that underpin the effect of miR-195, is currently **incomplete**.

---

## [Referee Report · Reviewer #1 (Public review)]

Summary:

Here, the authors are proposing a role for miR-196, a microRNA that has been shown to bind and enhance degradation of mRNA targets in the regulation of cell processes, has a novel role in allowing the emergence of CD19+ cells in cells in which Ebf1, a critical B-cell transcription factor, has been genetically removed.

Strengths:

That over-expression of mR-195 can allow the emergence of CD19+ cells missing Ebf1 is somewhat novel.

Their data does perhaps support to a degree the emergence of a transcriptional network that may bypass the absence of Ebf1, including the FOXO1 transcription factor, but this data is not strong or definitive.

Weaknesses:

It is unclear whether this observation is in fact physiological. When the authors analyse a knockout model of miR-195, there is not much of a change in the B-cell phenotype. Their findings may therefore be an artefact of an overexpression system.

The authors have provided insufficient data to allow a thorough appraisal of the step-wise molecular changes that could account for their observed phenotype.

On review of the resubmitted manuscript, while I note the authors have attempted to address several of my comments, unfortunately, their resubmission is not sufficient to address several of the comments I had previously made.

In particular, in the resubmitted data that includes western blots for PAX5 and ERG in their EBF1-/- model, Supp Fig S3, the bands they show infer that that PAX5 and ERG expression can still be significantly detected in their EBF1-/- early B-cell model. This should not be the case, as no expression of PAX5 or ERG should be seen, as has been shown in prior literature.

---

## [Referee Report · Reviewer #2 (Public review)]

Summary:

The authors investigate miRNA miR-195 in the context of B-cell development. They demonstrate that ectopic expression of miR-195 in hematopoietic progenitor cells can, to a considerable extent, override the consequences of deletion of Ebf1, a central B-lineage defining transcription factor, in vitro and upon short-term transplantation into immunodeficient mice in vivo. In addition, the authors demonstrate that the reverse experiment, genetic deletion of miR-195, has virtually no effect on B-cell development. Mechanistically, the authors identify Foxo1 phosphorylation as one pathway partially contributing to the rescue effect of miR-195. An additional analysis of epigenetics by ATACseq adds potential additional factors that might also contribute to the effect of ectopic expression of miR-195.

Strengths:

The authors employ a robust assay system, Ebf1-KO HPC, to test for B-lineage promoting factors. The manuscript overall takes on an interesting perspective rarely employed for analysis of miRNA by overexpressing the miRNA of interest. Ideally, this approach may reveal, if not the physiological function of this miRNA, the role of distinct pathways in developmental processes.

Weaknesses:

At the same time, this approach constitutes a major weakness: It does not reveal information on the physiological role of miR-195. In fact, the authors themselves demonstrate in their KO approach, that miR-195 has virtually no role in B-cell development, as has been demonstrated already in 2020 by Hutter and colleagues. While the authors cite this paper, unfortunately, they do so in a different context, hence omitting that their findings are not original.

Conceptually, the authors stress that a predominant function of miRNA (in contrast to transcription factors, as the authors suggest) lies in fine-tuning. However, there appears to be a misconception. Misregulation of fine tuning of gene expression may result in substantial biological effects, especially in developmental processes. The authors want to highlight that miR-195 is somewhat an exception in that regard, but this is clearly not the case. In addition to miR-150, as referenced by the authors, also the miR-17-92 or miR-221/222 families play a significant role in B-cell development, their absence resulting in stage-specific developmental blocks, and other miRNAs, such as miR-155, miR-142, miR-181, and miR-223 are critical regulators of leukocyte development and function. Thus, while in many instances a single miRNA moderately affects gene expression at the level of an individual target, quite frequently targets converge in common pathways, hence controlling critical biological processes.

The paper has some methodological weaknesses as well: For the most part, it lacks thorough statistical analysis and only representative FACS plots are provided. Many bar graphs are based on heavy normalization making the T-tests employed inapplicable. No details are provided regarding statistical analysis of microarrays. Generation of the miR-195-KO mice is insufficiently described and no validation of deletion is provided. Important controls are missing as well, the most important one being a direct rescue of Ebf1-KO cells by re-expression of Ebf1. This control is critical to quantify the extent of override of Ebf1-deficiency elicited by miR-195 and should essentially be included in all experiments. A quantitative comparison is essential to support the authors' main conclusion highlighted in the title of the manuscript. As the manuscript currently stands, only negative controls are provided, which, given the profound role of Ebf1, are insufficient, because many experiments, such as assessment of V(D)J recombination, IgM surface expression, or class-switch recombination, are completely negative in controls. In addition, the authors should also perform long-term reconstitution experiments. While it is somewhat surprising that the authors obtain splenic IgM+ B cells after just 10 days, these experiments would certainly be much more informative after longer periods of time. Using "classical" mixed bone marrow chimeras using a combination of B-cell defective (such as mb1/mb1) bone marrow and reconstituted Ebf1-KO progenitors would permit much more refined analyses.

With regard to mechanism, the authors show that the Foxo1 phosphorylation pathway accounts for the rescue of CD19 expression, but not of other factors, and mentioned in the discussion. The authors then resort to epigenetic analysis, but their rationale remains somewhat vague. It remains unclear how miR-195 is linked to epigenetic changes.

---

## [Referee Report · Reviewer #3 (Public review)]

Summary:

In this study, Miyatake et al. present the interesting finding that ectopic expression of miR-195 in EBF1-deficient hematopoietic progenitor cells can partially rescue their developmental block and allows B cells to progress to a B220+ CD19+ cells stage. Notably, this is accompanied by an upregulation of B cell specific genes and, correspondingly, a downregulation of T, myeloid and NK lineage-related genes, suggesting that miR-195 expression is at least in part equivalent to EBF1 activity in orchestrating the complex gene regulatory network underlying B cell development. Strengthening this point, ATAC sequencing of miR-195-expressing EBF1-deficient B220+CD19+ cells and a comparison of these data to public datasets of EBF1-deficient and -proficient cells suggest that miR-195 indirectly regulates gene expression and chromatin accessibility of some, but not all regions regulated by EBF1.

Mechanistically, the authors identify a subset of potential target genes of miR-195 involved in MAPK and PI3K signalling. Dampening of these pathways has previously been demonstrated to activate FOXO1, a key transcription factor for early B cells downstream of EBF1. Accordingly, the authors hypothesize that miR-195 exerts its function through FOXO1. Supporting this claim, also exogenous FOXO1 expression is able to promote the development of EBF1-deficient cells to the B220+CD19+ stage and thus recapitulates the miR-195 phenotype.

Strengths:

The strength of the presented study is the detailed assessment of the altered chromatin accessibility in response to ectopic miR-195 expression. This provides insight into how miR-195 impacts on the gene regulatory network that governs B cell development and allows the formation of mechanistic hypotheses.

Weaknesses:

The key weakness of this study is that its findings are based on the artificial and ectopic expression of a miRNA out of its normal context, which in my opinion strongly limits the biological relevance of the presented work. While the authors performed qPCRs for miR-195 on different B cell populations and show that its relative expression peaks in early B cells, it remains unclear whether the absolute miR-195 expression is sufficiently high to have any meaningful biological activity. In fact, other miRNA expression data from immune cells (e.g. DOI 10.1182/blood-2010-10-316034 and DOI 10.1016/j.immuni.2010.05.009) suggest that miR-195 is only weakly, if at all, expressed in the hematopoietic system.

Update to this part after revision: The authors now state in the discussion that their study does not aim to uncover and characterize a physiological role of miR-195 in lymphocytes development, but rather reveals "the potential of miR-195 to compensate for EBF1 deficiency". However, in my opinion, the absence of any physiological context still limits this study's relevance.

The authors support their finding by a CRISPR-derived miR-195 knockout mouse model which displays mild but significant differences in the hematopoietic stem cell compartment and in B cell development. However, they fail to acknowledge and discuss a lymphocyte-specific miR-195 knockout mouse that does not show any B cell defects in the bone marrow or spleen and thus contradicts the authors' findings (DOI 10.1111/febs.15493). Of note, B-1 B cells in particular have been shown to be elevated upon loss of miR-15-16-1 and/or miR-15b-16-2, which contradicts the data presented here for loss of the family member miR-195.

A second weakness is that some claims by the authors appear overstated or at least not fully backed up by the presented data. In particular, the findings that miR-195-expressing cells can undergo VDJ recombination, express the pre-BCR/BCR and can class switch need to be strengthened. It would be beneficial to include additional controls to these experiments, e.g. a RAG-deficient mouse as a reference/negative control for the ddPCR and the surface IgM staining, and cells deficient in class switching for the IgG1 flow cytometric staining.

Moreover, the manuscript would be strengthened by a more thorough investigation of the hypothesis that miR-195 promotes the stabilization and activity of FOXO1, e.g. by comparing the authors' ATACseq data to the FOXO1 signature.

---

## [Author Response]

The following is the authors’ response to the original reviews.

**eLife assessment**
This useful study reports that the exogenous expression of the microRNA miR-195 can partially compensate in early B cell development for the loss of EBF1, one of the key transcription factors in B cells. While this finding will be of interest to those studying lymphocyte development, the evidence, particularly with regard to the molecular mechanisms that underpin the effect of miR-195, is currently incomplete.
**Public Reviews:**

**Reviewer #1 (Public review):**
Summary:Here, the authors are proposing a role for miR-196, a microRNA that has been shown to bind and enhance the degradation of mRNA targets in the regulation of cell processes, and has a novel role in allowing the emergence of CD19+ cells in cells in which Ebf1, a critical B-cell transcription factor, has been genetically removed.Strengths:That over-expression of mR-195 can allow the emergence of CD19+ cells missing Ebf1 is somewhat novel.Their data does perhaps support to a degree the emergence of a transcriptional network that may bypass the absence of Ebf1, including the FOXO1 transcription factor, but this data is not strong or definitive.Weaknesses:It is unclear whether this observation is in fact physiological. When the authors analyse a knockout model of miR-195, there is not much of a change in the B-cell phenotype. Their findings may therefore be an artefact of an overexpression system.The authors have provided insufficient data to allow a thorough appraisal of the stepwise molecular changes that could account for their observed phenotype.
**Reviewer #2 (Public review):**
Summary:The authors investigate miRNA miR-195 in the context of B-cell development. They demonstrate that ectopic expression of miR-195 in hematopoietic progenitor cells can, to a considerable extent, override the consequences of deletion of Ebf1, a central Blineage defining transcription factor, in vitro and upon short-term transplantation into immunodeficient mice in vivo. In addition, the authors demonstrate that the reverse experiment, genetic deletion of miR-195, has virtually no effect on B-cell development. Mechanistically, the authors identify Foxo1 phosphorylation as one pathway partially contributing to the rescue effect of miR-195. An additional analysis of epigenetics by ATACseq adds potential additional factors that might also contribute to the effect of ectopic expression of miR-195.Strengths:The authors employ a robust assay system, Ebf1-KO HPC, to test for B-lineage promoting factors. The manuscript overall takes on an interesting perspective rarely employed for the analysis of miRNA by overexpressing the miRNA of interest. Ideally, this approach may reveal, if not the physiological function of this miRNA, the role of distinct pathways in developmental processes.Weaknesses:At the same time, this approach constitutes a major weakness: It does not reveal information on the physiological role of miR-195. In fact, the authors themselves demonstrate in their KO approach, that miR-195 has virtually no role in B-cell development, as has been demonstrated already in 2020 by Hutter and colleagues. While the authors cite this paper, unfortunately, they do so in a different context, hence omitting that their findings are not original.Conceptually, the authors stress that a predominant function of miRNA (in contrast to transcription factors, as the authors suggest) lies in fine-tuning. However, there appears to be a misconception. Misregulation of fine-tuning of gene expression may result in substantial biological effects, especially in developmental processes. The authors want to highlight that miR-195 is somewhat of an exception in that regard, but this is clearly not the case. In addition to miR-150, as referenced by the authors, also the miR-17-92 or miR-221/222 families play a significant role in B-cell development, their absence resulting in stage-specific developmental blocks, and other miRNAs, such as miR-155, miR-142, miR-181, and miR-223 are critical regulators of leukocyte development and function. Thus, while in many instances a single miRNA moderately affects gene expression at the level of an individual target, quite frequently targets converge in common pathways, hence controlling critical biological processes.The paper has some methodological weaknesses as well: For the most part, it lacks thorough statistical analysis, and only representative FACS plots are provided. Many bar graphs are based on heavy normalization making the T-tests employed inapplicable. No details are provided regarding the statistical analysis of microarrays. Generation of the miR-195-KO mice is insufficiently described and no validation of deletion is provided. Important controls are missing as well, the most important one being a direct rescue of Ebf1-KO cells by re-expression of Ebf1. This control is critical to quantify the extent of override of Ebf1-deficiency elicited by miR-195 and should essentially be included in all experiments. A quantitative comparison is essential to support the authors' main conclusion highlighted in the title of the manuscript. As the manuscript currently stands, only negative controls are provided, which, given the profound role of Ebf1, are insufficient, because many experiments, such as assessment of V(D)J recombination, IgM surface expression, or class-switch recombination, are completely negative in controls. In addition, the authors should also perform long-term reconstitution experiments. While it is somewhat surprising that the authors obtained splenic IgM+ B cells after just 10 days, these experiments would be certainly much more informative after longer periods of time. Using "classical" mixed bone marrow chimeras using a combination of B-cell defective (such as mb1/mb1) bone marrow and reconstituted Ebf1-KO progenitors would permit much more refined analyses.With regard to mechanism, the authors show that the Foxo1 phosphorylation pathway accounts for the rescue of CD19 expression, but not for other factors, as mentioned in the discussion. The authors then resort to epigenetics analysis, but their rationale remains somewhat vague. It remains unclear how miR-195 is linked to epigenetic changes.
**Reviewer #3 (Public review):**
Summary:In this study, Miyatake et al. present the interesting finding that ectopic expression of miR-195 in EBF1-deficient hematopoietic progenitor cells can partially rescue their developmental block and allow B cells to progress to a B220+ CD19+ cells stage. Notably, this is accompanied by an upregulation of B-cell-specific genes and, correspondingly, a downregulation of T, myeloid, and NK lineage-related genes, suggesting that miR-195 expression is at least in part equivalent to EBF1 activity in orchestrating the complex gene regulatory network underlying B cell development. Strengthening this point, ATAC sequencing of miR-195-expressing EBF1-deficient B220+CD19+ cells and a comparison of these data to public datasets of EBF1-deficient and -proficient cells suggest that miR-195 indirectly regulates gene expression and chromatin accessibility of some, but not all regions regulated by EBF1.Mechanistically, the authors identify a subset of potential target genes of miR-195 involved in MAPK and PI3K signaling. Dampening of these pathways has previously been demonstrated to activate FOXO1, a key transcription factor for early B cells downstream of EBF1. Accordingly, the authors hypothesize that miR-195 exerts its function through FOXO1. Supporting this claim, also exogenous FOXO1 expression is able to promote the development of EBF1-deficient cells to the B220+CD19+ stage and thus recapitulates the miR-195 phenotype.Strengths:The strength of the presented study is the detailed assessment of the altered chromatin accessibility in response to ectopic miR-195 expression. This provides insight into how miR-195 impacts the gene regulatory network that governs B-cell development and allows the formation of mechanistic hypotheses.Weaknesses:The key weakness of this study is that its findings are based on the artificial and ectopic expression of a miRNA out of its normal context, which in my opinion strongly limits the biological relevance of the presented work.While the authors performed qPCRs for miR-195 on different B cell populations and show that its relative expression peaks in early B cells, it remains unclear whether the absolute miR-195 expression is sufficiently high to have any meaningful biological activity. In fact, other miRNA expression data from immune cells (e.g. DOI 10.1182/blood-2010-10-316034 and DOI 10.1016/j.immuni.2010.05.009) suggest that miR-195 is only weakly, if at all, expressed in the hematopoietic system.The authors support their finding by a CRISPR-derived miR-195 knockout mouse model which displays mild, but significant differences in the hematopoietic stem cell compartment and in B cell development. However, they fail to acknowledge and discuss a lymphocyte-specific miR-195 knockout mouse that does not show any B cell defects in the bone marrow or spleen and thus contradicts the authors' findings (DOI 10.1111/febs.15493). Of note, B-1 B cells in particular have been shown to be elevated upon loss of miR-15-16-1 and/or miR-15b-16-2, which contradicts the data presented here for loss of the family member miR-195.A second weakness is that some claims by the authors appear overstated or at least not fully backed up by the presented data. In particular, the findings that miR-195expressing cells can undergo VDJ recombination, express the pre-BCR/BCR and class switch needs to be strengthened. It would be beneficial to include additional controls to these experiments, e.g. a RAG-deficient mouse as a reference/negative control for the ddPCR and the surface IgM staining, and cells deficient in class switching for the IgG1 flow cytometric staining.Moreover, the manuscript would be strengthened by a more thorough investigation of the hypothesis that miR-195 promotes the stabilization and activity of FOXO1, e.g. by comparing the authors' ATACseq data to the FOXO1 signature.
**Recommendations for the authors:**

**Reviewer #1 (Recommendations for the authors):**
Miyatake et al., present a manuscript that explores the role of miR-195 in B cell development.Their data suggests a role for this microRNA:Using an Ebf1 fetal liver knockout of B-cell differentiation that a small population of CD19 expressing with some evidence of V(D)J recombination capable of class switch can be derived by transduction of miR-195.In the emergent CD19+ Ebf1-/- cells, the authors provide some evidence that Mapk and Akt3 may be miR-195 targets that are downregulated allowing FOXO1 transcription factor pathway may be involved in the emergent CD19+ cells arising from miR-195 transduction.Perhaps less compelling data is provided with regards to a role for miR-195 in normal Bcell development through analysis of a miR-195 knockout model.While there are some interesting preliminary data presented for a role for miR-195 in the context of Ebf1-/- cells, there are some questions I think the authors could consider.Comments:(1-1) It is difficult to ascertain the potential role of miR-195 transduction in allowing the emergence of CD19+ cells from the data provided. miR-195 has been generally shown to destabilize mRNA transcripts by 3' UTR binding that targets mRNA transcripts for degradation. The effect of transduction of miR-195 would therefore be expected to be related to the degradation of factors opposing aspects of B-lineage specification or maintenance. I would be particularly interested in transcriptional or epigenetic regulators that may be modified in this way, at an mRNA as well as protein level.

We appreciate the reviewerʼs thoughtful comments and agree that miRNAs often exert their effects through the degradation or translational repression of mRNAs encoding regulatory factors. In our study, we attempted to address this point by combining predictive analysis (using TargetScan and starBase) with luciferase reporter assays and qPCR to validate several potential targets of miR-195, including Mapk3 and Akt3. We acknowledge that this is not a comprehensive mechanistic analysis. We agree that a broader and systematic identification of direct targets of miR-195, particularly those involved in transcriptional and epigenetic regulation, would further clarify the mechanisms involved. However, due to limitations in resources and time, we are currently unable to perform global proteomic or ChIP-based validations. Nevertheless, our ATAC-seq and microarray data indicate that miR-195 overexpression leads to increased accessibility and expression of several key B-lineage transcription factors (Pax5, Runx1, Irf8), suggesting that miR-195 indirectly activates transcriptional programs relevant to B cell commitment. We have now clarified this limitation in the revised Discussion section (lines 505‒524), and we emphasize that our current findings represent the potential of miR-195 rather than its physiological role. We hope that this clarification addresses the concern.

(1-2) While I acknowledge the authors have undertaken TargetScan and starBase analysis to try and predict miR-195 interactions, they do not provide a comprehensive list of putative targets that can be referenced against their cDNA data. Though they postulate Mapk3 and Akt3 as putative miR-195 targets and assay these in luciferase reporter systems (Figure 4), these were not clearly differentially regulated in the microarray data they provided (Figure 1E) as being downregulated on miR-195 transduction in Ebf1-/- cells.

We thank the reviewer for pointing out the need for a more comprehensive list of predicted miR-195 targets. In response, we have now included a supplementary table 4 (human) and 5 (mouse) listing all putative miR-195 targets predicted by TargetScan and starBase. As noted, Mapk3 expression was indeed downregulated upon miR-195 transduction, consistent with our luciferase reporter and qPCR results. For Akt3, we observed variability in the microarray data depending on the probe used, resulting in inconsistent expression levels. We acknowledge this and have added a clarification in the revised manuscript (lines 335‒339), noting that the regulation of Akt3 by miR-195 is potentially probe-dependent and may require further validation. We hope this clarification resolves the concern.

(1-3) The authors should provide a more comprehensive analysis of transcriptional changes induced by miR-195 Ebf1-/- specifically in the preproB cell stage of development in Ebf1-/- and miR-195 Ebf1-/- cells. The differentially expressed gene list should be provided as a supplemental file. The gene expression data should be provided for the different B-cell differentiation stages, eg. Ebf1-/- preproB cells, and Ebf1-/- miR-195 preproB cells, CD19+ cells and more differentiated subsets induced by miR-195 transduction.

We appreciate the reviewerʼs suggestion to provide a more comprehensive transcriptomic analysis at different B-cell differentiation stages. Unfortunately, due to the limited availability of cells and technical constraints, we were unable to perform RNA-seq on miR-195 transduced Ebf1^−/−^ pre-pro-B or CD19+ cells. However, to address this point, we referenced publicly available RNA-seq data (GEO accession: GSE92434), which includes transcriptomic profiles of Ebf1^−/−^ pro-B cells and wild-type controls. By comparing our microarray data from miR-195 transduced Ebf1^−/−^ cells with this dataset, we found partial restoration of expression for several key B-lineage genes, such as Pax5, Runx1, and Irf8, which are normally downregulated in the absence of EBF1. This comparison supports the notion that miR-195 partially reactivates the transcriptional network essential for B cell development. We have added this interpretation to the Discussion section (lines 528‒533).

(1-4) More replicates (at least 3 of each genotype) are required for their Western Blots for FOXO1 and pFOXO1 (Fig 4C, D). Western blots should also be provided for other known B-lineage transcriptional regulators such as PAX5 and ERG.

We thank the reviewer for these valuable suggestions. In response, we have now quantified and added the relative band intensities of FOXO1 and pFOXO1 from three independent experiments in the revised Figure 4C, and we include statistical analysis to support the reproducibility of these results. Additionally, as requested, we performed western blotting for PAX5 and ERG using the same samples. The results showed no significant change in these protein levels between miR-195-transduced and control Ebf1^−/−^ cells, consistent with the modest upregulation observed in our microarray data. We have included the PAX5 and ERG western blot images in Supplementary Figure S3 and have revised the text in the Results section (lines 351‒35)

(1-5) The authors have not shown a transcriptional binding by ChIPseq or other methods such as cut and tag/ cut and run for FOXO1 binding to B-lineage genes in their Ebf1-/- miR-195 CD19+ cells to be able to definitively show this TF is critical for the emergence of the C19+ cell phenotype by demonstrating direct binding to "upregulated" genes cis-regulatory regions in the Ebf1-/- miR-195 CD19+ cells

We appreciate the reviewerʼs suggestion regarding the use of ChIP-seq or related methods to demonstrate direct FOXO1 binding to cis-regulatory regions of B-lineage genes in Ebf1^−/−^ miR-195 CD19⁺ cells. We agree that such data would provide definitive evidence of FOXO1's direct involvement in promoting the B cell-like transcriptional program. However, due to current technical limitations, including the scarcity of CD19⁺ cells derived from Ebf1^−/−^ miR-195 transduction and the requirement for large cell numbers in ChIP-seq or CUT&RUN protocols, we were unable to perform these assays in this study. Nevertheless, our current data provide multiple lines of indirect evidence supporting the involvement of FOXO1:

miR-195 transduction leads to reduced phosphorylation and increased accumulation of FOXO1 protein (Fig. 4C).

Overexpression of FOXO1 in Ebf1^−/−^ HPCs partially recapitulates the miR-195 phenotype (Fig. 4D).

ATAC-seq data show increased chromatin accessibility at known FOXO1 target gene loci (e.g., Pax5, Runx1, Irf8) in miR-195-induced CD19⁺ cells, many of which overlap with FOXO1 motifs(Fig.5)

These observations collectively suggest that FOXO1 activity is functionally important for the emergence of CD19⁺ cells, even though direct binding has not been confirmed. We have added this limitation to the Discussion (lines 531‒537), and we note that future studies using FOXO1 CUT&RUN in this system would be valuable to further define the underlying mechanism.

(1-6) The authors have not shown significant upregulation of expression of other critical B-cell regulatory transcription factors in their Ebf1-/- miR-195 CD19+ cells that could account for the emergence of these cells such as Pax5 or Erg. The legend in Figure 1E suggests for example the change in expression of Pax5 is modest if anything at best as no LogFC or western blot data is presented.

We thank the reviewer for raising this point. In our microarray analysis (Figure 1D, original Figure 1E), we observed that both Pax5 and Erg mRNA levels were upregulated in Ebf1^−/−^ cells upon miR-195 transduction. Specifically, Pax5 showed an increase of approximately log₂FC 1.2, and Erg was also consistently elevated across biological replicates. These changes, although modest, were statistically significant and consistent with the upregulation of other B-lineage-associated transcription factors, such as Runx1 and Irf8. We agree that the magnitude of Pax5 upregulation is not as high as typically seen during full B cell commitment, and therefore may not have been immediately apparent in Figure 1D (original Figure 1E). To clarify this point, we have now revised the text in the Results section (lines 170‒174) to highlight the observed changes in Pax5 and Erg expression. We believe that the upregulation of these transcription factors, together with increased FOXO1 activity and changes in chromatin accessibility (Figure 5), contributes to the partial reactivation of the B cell gene regulatory network in the absence of EBF1.

(1-7) Which V(D)J transcripts have been produced? A more detailed analysis other than ddPCR is required to help understand the emergence of this population that can presumably proceed through the preBCR and BCR checkpoints.

We appreciate the reviewerʼs interest in understanding the nature of the V(D)J rearrangements in Ebf1^−/−^ miR-195 CD19⁺ cells. As noted, our current data rely on droplet digital PCR (ddPCR), which was used to detect rearranged VH-JH segments in the bone marrow of engrafted mice. While this approach does not allow for detailed mapping of specific V, D, or J gene usage, it provides a sensitive and quantitative measure of V(D)J recombination activity. The detection of rearranged VH-JH fragments in miR-195-transduced Ebf1^−/−^ cells suggests that at least partial recombination of the immunoglobulin heavy chain locus is occurring̶an essential checkpoint for progression past the pro-B cell stage. Given the lack of such rearrangements in control-transduced Ebf1^−/−^ cells, we interpret this as evidence that miR-195 enables cells to initiate the recombination process. We acknowledge the limitations of ddPCR and agree that a more detailed analysis using VDJ-seq or singlecell RNA-seq would be valuable in determining the diversity and completeness of the V(D)J transcripts produced. This is a direction we intend to pursue in future work. We have added this limitation to the Discussion section (lines 538‒543).

(1-8) The authors reveal that the Foxo1 transduced Ebf1-/- cells (Fig. 4D) do not persist in vitro or be detected via transplant assay (line 256) and therefore does not represent a truly "rescued" B cell, suggesting that CD19+ cells Ebf1-/- miR-195 transduced cells have more B-cell potential. Further characterisation is therefore warranted of this cell population. For instance, can these cells be induced to undergo myeloid differentiation in myeloid cytokine conditions? What other B-lineage transcriptional regulators are expressed in this cell population that could account for VDJ recombination and expression of a B-lineage transcriptional program (see comments 1, 3, and 5) that allow transition through preBCR and BCR checkpoints as well as undergo class switching?

We thank the reviewer for this insightful comment. We agree that the persistence and lineage potential of the CD19⁺ cells emerging from Ebf1^−/−^ miR-195-transduced progenitors deserve further characterization. Although we were unable to perform additional lineage re-direction assays, our current data provide several lines of evidence suggesting that these cells are stably committed toward the B-lineage:

Gene expression profiling revealed upregulation of multiple B cell transcriptional regulators, including Pax5, Runx1, and Irf8.

ATAC-seq analysis showed increased chromatin accessibility at B cell‒specific loci and enrichment of motifs bound by key B-lineage factors such as FOXO1 and E2A.

The cells express surface IgM and undergo class switch recombination to IgG1 upon stimulation, indicating successful transition through the pre-BCR and BCR checkpoints and acquisition of mature B cell functions.

Importantly, no upregulation of myeloid- or T-lineage genes was detected in the microarray analysis, arguing against multipotency at this stage.We acknowledge that functional tests for lineage plasticity under altered cytokine conditions would provide important insights and plan to address this question in future studies. This limitation has now been noted in the revised Discussion (lines 544‒550).

(1-9) In the original Ebf1-/- miR-195 CD19+ experiments, a wild-type control should be provided for each experiment.

We appreciate the reviewerʼs suggestion to include wild-type controls in all experiments. While we did not include wild-type samples side-by-side in every assay, we carefully designed our experiments to include biologically appropriate and informative comparisons. For example, in the bone marrow transplantation experiments (Figure 2), Ebf1^−/−^ cells transduced with empty vector served as negative controls, clearly lacking CD19 expression, V(D)J recombination, IgM surface expression, and class switch capability. This allowed us to specifically assess the gain-of-function effects of miR-195 in the EBF1-deficient background. In several analyses̶such as the ATAC-seq and microarray comparisons̶we did incorporate or refer to existing wild-type datasets (e.g., GSE92434), providing context for the extent of recovery toward a WT-like profile. We agree, however, that including parallel WT controls across all experimental platforms would enhance interpretability.

(1-10) For ATACseq data, a comparison between Ebf1-/- preproB cells and Ebf1-/- miR-195 CD19+ cells should be undertaken.

We thank the reviewer for this important point. As suggested, we have performed a direct comparison of chromatin accessibility between Ebf1_−/−_ pre-pro-B‒like cells (CD19_-_, control transduction) and Ebf1_−/−_ miR-195‒transduced CD19⁺ cells. This comparison is shown in green in Figure 5B and represents the ATAC-seq peaks differentially accessible between these two populations.

(1-11) I cannot agree with the authors with some of their statements such as Line 242 - "therefore miR-195 considered to have similar function with EBF1 to some extent" - how can this be the case when miR-195 is a miRNA and EBF1 is a transcription factor with pioneering transcriptional activity? Surely the effects of miR-195 must be secondary.

We thank the reviewer for pointing out the inappropriateness of comparing miR-195 to EBF1 in terms of functional similarity. We agree that miR-195, as a microRNA, operates through post-transcriptional regulation and does not possess the pioneering transcriptional activity characteristic of EBF1. To avoid confusion or overstatement, we have removed the sentence in line 242 ("therefore miR-195 is considered to have similar function with EBF1 to some extent").

(1-12) It is unclear whether this observation is in fact physiological. When the authors analyse a knockout model of miR-195, there is not much of a change in the B-cell phenotype. Their findings may therefore be an artefact of an overexpression system. The authors should comment on this observation in their discussion.

We thank the reviewer for this important observation. We agree that the mild phenotype observed in our miR-195 knockout mice suggests that miR-195 is not essential for B cell development under steady-state physiological conditions. Accordingly, we do not claim a physiological requirement for miR-195. Rather, our study demonstrates that miR-195 possesses the potential to activate a B-lineage program in the absence of EBF1 when ectopically expressed. This functional potential̶rather than its endogenous necessity̶ is the main focus of our work. We have now clarified this distinction in the revised Discussion section (lines 551‒560), and we emphasize that our findings highlight an alternative regulatory pathway that can be artificially engaged under specific conditions.

(1-13) I recommend the authors check spelling and grammar throughout their manuscript.

We thank the reviewer for the suggestion. In response, we have carefully reviewed the manuscript for spelling, grammar, and clarity. Minor corrections have been made throughout the text to improve readability and ensure consistency. We hope that the revised version addresses any language-related concerns. In addition, the manuscript has been reviewed by professional editing service to improve the language quality.

(1-14) In general, I recommend more comprehensive primary data be presented in the manuscript or supplementary files to add value to their submission.

We thank the reviewer for this helpful suggestion. In response, we have revised the manuscript and supplementary materials to include additional primary data wherever possible. The bar graphs have been updated to include individual data points to show variability and replicate information. Uncropped western blot images are now provided in Supplementary Figure S2. We hope these additions provide greater transparency and value to the manuscript.

**Reviewer #2 (Recommendations for the authors):**
I have a number of suggestions with regard to inclusion of details and controls:(2-1) The authors need to provide more details on in vitro differentiation, especially culture times.

Thank you for your comment. The culture conditions for in vitro differentiation of Ebf1^−/−^ hematopoietic progenitor cells are described in the Methods section (lines 648‒ 649) under “Culture of lineage-negative (Lin‒) cells from the fetal liver.” As stated, cells were cultured more than 7 days under the specified conditions.

(2-2) In Figure 1E, the authors need to provide information on statistics (FDR or similar).

I thank the reviewer for the suggestion. In Figure 1D (Original Figure 1E) (the microarray analysis), only two biological replicates were available for each condition (n = 2 per group). Due to this limited sample size, we did not perform statistical testing, as the power would be insufficient to produce reliable p-values or adjusted FDRs. Instead, we focused on genes with consistent and biologically meaningful changes in expression, and presented representative examples based on fold change values.

(2-3) For in vivo experiments (Figure 2) the authors should comment on their use of two different recipient mouse strains despite very low n numbers. As described above, classical mixed BM chimeras would be much more informative. In these experiments, the authors should also show the formation of other lymphoid lineages. This would answer the question of whether miR-195 redirects cells to the B lineage. Most importantly, absolute numbers need to be provided, especially in conjunction with Ebf1 rescue as described above.

We thank the reviewer for the thoughtful and detailed suggestions regarding our in vivo experiments. Regarding the use of different recipient mouse strains, our initial intention was to perform the transplantations in BRG mice; however, due to facility restrictions and animal husbandry considerations, we had to switch to NOG mice. All in vivo experiments were performed with n = 3 per group, in accordance with ethical guidelines and efforts to minimize animal use while still ensuring reproducibility. With respect to the suggestion of mixed bone marrow chimeras, we agree that this approach can provide valuable information on lineage competitiveness. However, in our system, miR-195 confers only a very limited B cell developmental potential in Ebf1^−/−^ progenitors. In such a setting, the inclusion of wild-type competitor cells would overwhelmingly dominate the B cell compartment, likely masking any measurable effect of miR-195. Therefore, we opted to assess the gain-of-function potential of miR-195 in a noncompetitive setting. Regarding the assessment of other lymphoid lineages, we focused our analysis on the emergence of B-lineage cells, as the frequency of CD19⁺ cells induced by miR-195 is quite low. Given this low efficiency, we consider it unlikely that miR-195 significantly alters the development of non-B lineages, and thus did not observe substantial lineage diversion effects. Our aim was not to demonstrate lineage redirection, but rather to show that miR-195 can confer partial B cell potential in the absence of EBF1.

Finally, we acknowledge the importance of presenting absolute cell numbers. However, the cell number collected from the mice were so few that we did not get the reliable results, we described it in the manuscript. (lines 498-501)

(2-4) The statistics in Figure 3 are inadequate. No S.D. is provided for WT. How then was normalization performed? Student's T-test cannot be applied to ratios.

We thank the reviewer for highlighting the need for more appropriate statistical analysis. Due to considerable inter-batch variability in absolute measurements, we normalized the KO values to their paired WT counterparts from the same experimental batch. Specifically, for each replicate, we calculated the KO/WT ratio to control for batch-specific variation. We then applied a one-sample t-test (against a null hypothesis of ratio = 1) to determine statistical significance. We have now revised the figure to show individual ratio values for each replicate and updated the legend and Methods to clearly explain the statistical approach. We hope this addresses the concern and improves the clarity and rigor of the analysis.

(2-5) In Figure 4A, the authors should comment on the strong repression of the Akt3UTR.

We appreciate the reviewerʼs observation regarding the strong repression observed with the Akt3 3'UTR construct. Indeed, we also noted that luciferase activity was markedly reduced in the presence of the Akt3 3'UTR, even in cells transduced with a control vector. We hypothesize that the Akt3 3'UTR contains strong post-transcriptional regulatory elements̶such as AU-rich elements or binding sites for endogenous miRNAs or RNA-binding proteins̶which may suppress mRNA stability or translation independent of miR-195. Alternatively, the secondary structure or length of the UTR may inherently reduce luciferase expression. We have added this limitation to the Discussion section (lines 561‒569).

(2-6) The Western blot in Figure 4C is of insufficient quality. The authors need to provide unspliced versions of the bands including markers.

We thank the reviewer for this important comment. In response, we have included the unprocessed, full-length Western blot images corresponding to Figure 4C as Fig. S2. This provides a transparent view of the original data and addresses the concern about image cropping.

(2-7) The ATACseq experiment in Figure 5 is difficult to comprehend. A simpler design including Ebf1 rescue controls would clearly improve this part.

We thank the reviewer for this valuable feedback. We agree that the original presentation of the ATAC-seq data may have been difficult to interpret. To address this, we have included a clear interpretation of the overlapping regions in the revised figure legend (lines 1018-1022). We hope this improves the clarity of the data and facilitates understanding of the chromatin changes mediated by EBF1 and miR-195.

(2-8) The miR-195 KO mouse lacks validation (RT-PCR, genomic PCR) as well as a clear description of the deleted region and whether miR-497 is affected. In addition, the genetic background and number of backcrosses for the removal of potential off-target effects need to be mentioned.

We thank the reviewer for this important comment. The miR-195 knockout mouse was generated via CRISPR/Cas9, and Sanger sequencing confirmed a 628 bp deletion on chromosome 11 (GRCm38/mm10 chr11:70,234,425‒70,235,103). This deletion includes the entire miR-497 locus and part of the miR-195 precursor sequence. Although we do not show PCR gel images, the deletion was validated by sequencing, and the results are now clearly described in the revised Methods section (lines 607619). All transgenic mice in this study were backcrossed to the C57BL/6 background for at least eight generations.

(2-9) The manuscript requires extensive editing for language.

We appreciate the reviewerʼs comment. The manuscript has now been revised and professionally edited for language by a native English-speaking editor. We believe clarity and readability have been significantly improved.

**Reviewer #3 (Recommendations for the authors):**
(3-1) What is the expression level of miR-195 after viral overexpression? In Figure 4B, the authors show a 2.5-fold increase, but this appears very low for the experimental system (expression through the MDH1 retroviral construct) and the observed repressive effects (e.g. Figure 4A and B).

We thank the reviewer for this insightful comment. We agree that the apparent ~2.5fold increase in miR-195 levels (Figure 4B) may seem modest in the context of retroviral overexpression and the associated functional effects. However, due to the high sequence similarity within the miR-15/16/195/497 family, it is technically challenging to measure mature miR-195 levels with complete specificity. The baseline signal observed in control samples likely reflects cross-reactivity with endogenous miRNAs such as miR-497 or miR-16, which share similar seed sequences. Therefore, the reported fold-change may underestimate the true level of ectopic miR-195 expression. Despite this, we observed robust repression of validated targets (e.g., Mapk3, Akt3) in both qPCR and luciferase assays, indicating that functionally effective levels of miR-195 were achieved. We have now clarified this limitation and interpretation in the revised Results sections (lines 332‒335).

(3-2) In alignment with the transparency of the data, I would encourage the authors to display the individual data points for all bar graphs.

We thank the reviewer for this helpful suggestion. In the revised manuscript, we have updated bar graphs to include individual data points to increase transparency and allow better visualization of data variability. In the ddPCR experiments, we provided the raw data in Fig. S1 for full transparency. In Fig. 1A, we have confirmed miR-195 expression profiles using the deposit data which the reviewer suggested, but miR-195 expression was very lower than we expected. We also performed scRNA-seq using hematopoietic lineage cells in 8-week-old C57BL/6 mice, but we could not get the reproducibility of miR-195 expression profiles. Therefore, we determined that this is an artifact caused by the miR-195 probe used for qPCR, and deleted Fig. 1A.

(3-3) The references appear to be compromised. For example, the authors state that "The Ebf1−/+ mouse was originally generated by R. Grosschedl (39)" (line 297), but this is not the respective paper. Likewise, the knockout mouse was generated "based on the CRISPR/Cas9 system established by C. Gurumurthy (40)" (line 299), but he/she is not involved in the referenced study.

We thank the reviewer for pointing out the discrepancies in the reference citations. Upon revising the Methods section to integrate it with the main text, the reference numbering became misaligned. We have corrected the reference in the revised manuscript, and we thank the reviewer for bringing this to our attention.

(3-4) Given that the miRNA Taqman assays the authors used here have difficulties to discriminate closely related miRNAs such as e.g. miR-16 (highly expressed in the hematopoietic system) and miR-195, I would suggest that the authors test their qPCR in an appropriate setup, e.g. in their knockout mouse model. In this context, did the authors use another small RNA as a reference for the qPCR analysis? In the methods, only GAPDH is mentioned, but in my opinion, another RNA that uses the same stemloop-based cDNA synthesis protocol would be better suited.

We thank the reviewer for this valuable and technically insightful comment.

As correctly pointed out, TaqMan-based qPCR assays for miRNAs such as miR-195 can show cross-reactivity with closely related family members, particularly miR-16, which is abundantly expressed in hematopoietic cells. Indeed, due to this limitation, we do not treat the qPCR results shown in the original Figures 1A and 4B as definitive quantification of miR-195 expression. Rather, these data are used to provide a suggestion and a rough estimate of overexpression efficiency, while our core functional analyses rely on phenotypic and molecular outcomes such as target gene repression and lineage emergence. With this in mind, although we acknowledge that a small RNA reference based on the same stem-loop cDNA synthesis would offer a more compatible normalization in principle, the inherent variability and lack of absolute specificity in such assays also limits their interpretive value. Therefore, we used GAPDH as a normalization control for consistency with other qPCR analyses in the manuscript. We have now clarified this rationale and limitation in the revised Methods sections (lines 712‒716), and we thank the reviewer again for highlighting this important technical consideration.

(3-5) The Western blot data used to support the hypothesis that FOXO1 phosphorylation is reduced upon overexpression of miR-195 are not convincing. The authors should not crop everything but the band.

We thank the reviewer for the helpful comment. In response, we have now provided the full-length, uncropped Western blot images corresponding to Figure 4C, including both total FOXO1 and phospho-FOXO1 blots. These images are included in Fig. S2.